# Preclinical Evaluation of Nanoemulsion and Polymeric Nanocapsule Delivery Systems of 4-(Phenylselanyl)-2H-Chromen-2-One for Rheumatoid Arthritis and Comorbidities

**DOI:** 10.3390/ph18091379

**Published:** 2025-09-16

**Authors:** Caren Aline Ramson da Fonseca, Vinicius Costa Prado, Letícia Cruz, Vanessa Macedo Esteves da Rocha, Ana Paula Bonato Wille, Angélica Schiavom dos Reis, Jean Carlo Kazmierczak, Ricardo Frederico Schumacher, Ethel Antunes Wilhelm

**Affiliations:** 1Graduate Program in Biochemistry and Bioprospecting, Preclinical and Translational Research Group on Pain and Chronic Diseases, Federal University of Pelotas, Pelotas CEP 96010-900, RS, Brazil; carenramson@hotmail.com (C.A.R.d.F.); vnsmacedo@gmail.com (V.M.E.d.R.); ana_paulabonatowille@outlook.com (A.P.B.W.); ge_schiavon@hotmail.com (A.S.d.R.); 2Laboratory of Pharmaceutical Technology, Graduate Program in Pharmaceutical Sciences, Health Sciences Center, Federal University of Santa Maria, Santa Maria CEP 97105-900, RS, Brazil; vini132007@gmail.com (V.C.P.); leticia.cruz@ufsm.br (L.C.); 3Graduate Program in Chemistry, Chemistry Department, Federal University of Santa Maria, Santa Maria CEP 97105-900, RS, Brazil; jeankazmierczak@gmail.com (J.C.K.); ricardo.schumacher@ufsm.br (R.F.S.)

**Keywords:** arthritis, nanotechnology, affective behaviors, hyperalgesia, oxidative stress

## Abstract

**Background/Objectives:** Recognizing the current limitations in rheumatoid arthritis (RA) treatments, especially in managing pain and inflammation, there is an urgent need to develop and explore new therapeutic strategies. In this study, we devised two innovative approaches using nanotechnology for treating RA. We evaluated the effectiveness of compound 4-(phenylselanyl)-2H-chromen-2-one (4-PSCO) in three forms: free, as well as in nanoemulsified (4-PSCO NE) and nanoencapsulated (4-PSCO NC) formulations. **Methods:** Arthritis was induced in mice by intraplantar injection of Freund’s complete adjuvant (CFA; 0.1 mL). The 4-PSCO free, 4-PSCO NE, and 4-PSCO NC (1 mg/kg, orally) treatments were administered daily for 15 days. We assessed disease signs, symptoms, mechanical and thermal sensitivities, neurobehavioral deficits, and activities of myeloperoxidase (MPO), Na^+^, K^+^-ATPase, and acetylcholinesterase (AChE), as well as oxidative stress markers. **Results:** Our study demonstrates, for the first time, that both 4-PSCO NC and 4-PSCO NE inhibit the clinical signs of RA in mice, including inflammation. Moreover, both formulations alleviated pain and anxiety behaviors while restoring AChE activity and decreasing oxidative stress in the cerebral cortex. Notably, only the 4-PSCO NC treatment increased the time animals spent in the open arms of the elevated plus-maze. It lowered TBARS levels in the cerebral cortex, spinal cord, and paws, showcasing its advantages over the free 4-PSCO and 4-PSCO NE. **Conclusions:** These findings highlight the therapeutic potential of 4-PSCO, especially the polymeric nanocapsule, as a practical option for treating both the symptoms and underlying mechanisms of RA.

## 1. Introduction

The pharmacotherapeutic management of rheumatoid arthritis (RA) has traditionally involved using active molecules aimed at relieving symptoms and employing therapeutic approaches that suppress the autoimmune response to foster disease remission [1]. RA is described as a multifactorial and complex autoimmune disorder with a chronic inflammatory nature and an unknown cause. The main clinical signs include damage to the joint, cartilage, and bone tissues [1,2,3]. Multisystem inflammation, combined with irreversible damage to osteoarticular structures, is the primary feature of RA [4].

Epidemiological studies show that RA affects about 1% of the world’s population, with prevalence reaching up to 2% in certain areas [5,6]. Recent research estimates that, among people aged 20 to 54, the global incidence of RA was nearly 508,185 cases. Additionally, from 1990 to 2021, the incidence rate increased from 11.66 to 13.48 per 100,000 people, highlighting not only the high prevalence but also the clinical significance and growing impact of this disease [5].

Pain is reported as one of the most critical and debilitating factors, significantly affecting the quality of life and being the main barrier to the psychological well-being of patients with RA. The connection between pain and RA is not yet fully understood; however, this condition results from peripheral inflammation, nociception, and oxidative stress [7,8,9]. Even after the inflammatory process goes into remission, many patients still experience residual pain, which requires different treatment strategies [9]. In this context, recent findings in psychiatry suggest that inflammation plays a significant role in the development of mental disorders, including depression and anxiety [10,11]. Research indicates that symptoms of anxiety and depression are more pronounced in patients with RA than in the general population, and these symptoms are directly connected to the severity of pain experienced [12,13,14].

Given its complexity, RA requires a multidisciplinary approach that emphasizes early diagnosis and timely intervention [15,16]. Although current strategies, including disease-modifying antirheumatic drugs and glucocorticoids, are effective in controlling peripheral inflammation and immune dysregulation, they are insufficient to address neuroinflammation, oxidative imbalance within the central nervous system (CNS), or psychiatric comorbidities such as anxiety and depression, factors that substantially impair patients’ quality of life [13,14]. Additionally, these treatments are often linked to adverse outcomes like mood disturbances, insomnia, neurotransmitter dysregulation, and endocrine changes, which can ultimately worsen pain sensitivity and contribute to hyperalgesia [1,17,18]. This underscores a critical unmet need for therapies capable of targeting not only joint inflammation but also CNS-related dysfunctions.

In this context, our research group recently examined the molecular interaction of 4-(phenylselanyl)-2H-chromen-2-one (4-PSCO) with proteins involved in pain and inflammation, using molecular docking simulations. The results revealed a strong binding affinity of free 4-PSCO to key targets, including p38 MAP kinase, peptidyl arginine deiminase type 4, phosphoinositide 3-kinase, Janus kinase 2, toll-like receptor 4, and nuclear factor-κB. Additionally, 4-PSCO-loaded nanocapsule suspensions were created and tested in preclinical models of acute and inflammatory pain, showing sustained antinociceptive and anti-inflammatory effects. Toxicity tests in nematodes and mice further confirmed the compound’s safety profile, with no significant adverse effects observed.

Building on these findings, 4-PSCO, a compound with promising pharmacological properties, emerges as an attractive candidate for RA treatment. Although 4-PSCO has promising pharmacological features, its physicochemical properties, such as high lipophilicity, could reduce its bioavailability. Nanotechnology enables the development of pharmaceutical formulations with controlled drug release and improved bioavailability, especially for highly lipophilic compounds [19,20,21]. Therefore, this study investigates a new therapeutic approach for RA by evaluating 4-PSCO within two distinct nanostructured delivery systems: polymeric nanocapsules and nanoemulsions. Polymeric nanocapsules are made of nanoparticles with an oily core surrounded by a thin polymer shell, while nanoemulsions are oil droplets dispersed in an aqueous phase, stabilized by surfactants. In our previous study, we developed a polymeric nanocapsule using ethylcellulose, a hydrophobic cellulose derivative known for its ability to control drug release. Although nanoencapsulated 4-PSCO demonstrated better pharmacological performance than its free form in some parameters in acute inflammation models, we hypothesized that modifying the formulation’s composition could further improve its therapeutic effects compared to the free form. Consequently, this study focuses on developing two distinct nanostructured systems: a polymeric nanocapsule using Poly(ε-caprolactone) and a nanoemulsion.

## 2. Results

### 2.1. Nanocapsule Suspension and Nanoemulsion Physicochemical Characterization

The results of the physicochemical characterization of the formulations are shown in Table 1. Both formulations had an average diameter in the nanometric range, below 260 nm, with a narrow size distribution, as indicated by a PDI value of less than 0.2. Additionally, the formulations exhibited a negative zeta potential and a slightly acidic pH. The statistical analysis using an unpaired Student *t*-test revealed a significant difference in the zeta potential values of the nanoemulsion containing 4-PSCO (4-PSCO NE) compared to the placebo nanoemulsion (NE P) (** *p* < 0.01). For both formulations containing 4-PSCO (nanocapsule suspensions with 4-PSCO (4-PSCO NC) and 4-PSCO NE), the encapsulation efficiency of 4-PSCO was 99%, and the drug content was approximately 100%.

Figure 1 illustrates the release profiles of 4-PSCO from both nanocarrier dispersions. Based on the release profiles observed over 12 h, nanocapsules more effectively control the release of 4-PSCO than the nanoemulsions. Additionally, it is noteworthy that the nanoemulsion exhibited signs of instability, such as phase separation, within 12 h, necessitating the interruption of the experiment for this sample after that period. However, the nanocapsules remained stable and released 62% of 4-PSCO over a 48-h period.

### 2.2. Therapeutic Effects of 4-PSCO in Free or Nanoencapsulated Form on the Arthritis Severity of Arthritis in CFA-Induced RA

The arthritis scores for female mice exposed to CFA are shown in Figure 2A,B. The results show that animals exposed to CFA induction exhibited high arthritis scores. Notably, daily treatment with 4-PSCO free, 4-PSCO NE, and 4-PSCO NC effectively reduced arthritis symptoms, including swelling, edema, and erythema in the ipsilateral paw of mice exposed to CFA, compared with the CFA, CFA+NE P, and CFA+NC P groups (Figure 2A,B) (ANOVA: F _(6, 42)_ = 17.04, *p* < 0.0001). No significant differences were observed between the CFA+4-PSCO free, CFA+4-PSCO NE, and CFA+4-PSCO NC groups.

### 2.3. Mitigation of RA Symptoms Through the Anti-Inflammatory Effects of 4-PSCO in Free or Nanoencapsulated Form

As shown in Figure 3, there was a significant increase in paw diameter (Figure 3A) and circumference (Figure 3B) in the CFA, CFA+NE P, and CFA+NC P groups compared to the Control, NE P, and NC P groups, respectively. Specifically, paw diameter increased by 33%, 34%, and 36% in these groups. For paw circumference, the increases of 25%, 25%, and 22%, respectively, were observed compared to the Control, NE P, and NC P groups. Notably, treatments with 4-PSCO free, 4-PSCO NE, and 4-PSCO NC (1 mg kg^−1^, i.g.) effectively reduced paw diameter increases compared to the CFA group (ANOVA: F _(6, 42)_ = 18.86, *p* < 0.0001) (Figure 3A). However, only the 4-PSCO free and 4-PSCO NC treatments significantly decreased CFA-induced paw swelling in terms of circumference (Figure 3B) (ANOVA: F _(6, 42)_ = 11.24, *p* < 0.0001). No significant differences were seen between the CFA+4-PSCO free, CFA+4-PSCO NE, and CFA+4-PSCO NC groups. The detailed group comparisons are available in the Appendix A.

Similarly, mice exposed to CFA showed a significant increase in paw edema compared to the Control group. Conversely, treatment with 4-PSCO free, 4-PSCO NE, and 4-PSCO NC (1 mg kg^−1^, i.g.) effectively reduced the paw edema in CFA-exposed animals (ANOVA: F _(6, 42)_ = 24.85, *p* < 0.0001). No statistical differences were observed among the CFA+4-PSCO free, CFA+4-PSCO NE, and CFA+4-PSCO NC groups (Figure 4A).

To confirm the increased inflammatory response caused by CFA, we measured MPO activity. As shown in Figure 4B, CFA elevated MPO activity in the paws of mice in the CFA group compared to the Control group. Data analysis revealed that treatment with 4-PSCO NC restored MPO enzyme activity in the paws of mice with CFA-induced RA (ANOVA: F _(6, 42)_ = 3.775, *p* < 0.01) (Figure 4B). However, the 4-PSCO free and 4-PSCO NE treatments did not significantly reduce the CFA-induced MPO activity. No statistical difference was observed between the CFA+4-PSCO free, CFA+4-PSCO NE, and CFA+4-PSCO NC groups regarding MPO activity in the paws of mice. The comparison between the groups, per se, is presented in the Appendix A.

### 2.4. Administration of 4-PSCO in Free or Nanoencapsulated Form Attenuates the Mechanical and Thermal Nociception Induced by CFA in Mice

The results compiled in this section demonstrate that treatments with 4-PSCO free, 4-PSCO NE, and 4-PSCO NC attenuated the nociceptive signs induced by CFA in mice (Figure 5). Before CFA administration, baseline measurements of paw withdrawal thresholds to mechanical stimuli were comparable across all experimental groups. Following CFA exposure, mice exhibited a marked decrease in paw withdrawal thresholds compared to control animals, which began on day 5 and persisted through at least day 10 of the study period.

On day 5, approximately a 60% reduction in paw withdrawal threshold was observed in the CFA, CFA+NE P, CFA+NC P, CFA+4-PSCO NE, and CFA+4-PSCO NC groups compared to the Control group (Figure 6). Treatment with 4-PSCO free, 4-PSCO NE, and 4-PSCO NC increased the paw withdrawal threshold in mice exposed to CFA, indicating that this compound and its formulations reversed the mechanical sensitivity caused by the AR model. Notably, on day 15, 4-PSCO free, 4-PSCO NE, and 4-PSCO NC resulted in increases of 66%, 80%, and 86%, respectively, in the paw withdrawal threshold in mice exposed to CFA (ANOVA: F _(6, 42)_ = 298.2, *p* < 0.0001). It is important to note that treatments with 4-PSCO NE and 4-PSCO NC were more effective at reducing sensitivity to mechanical stimuli than 4-PSCO free (Figure 5). A comparison of the groups is presented in the Appendix A.

As illustrated in Figure 6, the baseline values of paw withdrawal latency to thermal stimulus were equivalent among all experimental groups before the CFA exposure. On day 5, the animals exposed to CFA displayed a significant decrease in paw withdrawal latency, and the thermal sensitivity was sustained in the CFA groups until the last day of the experimental protocol. On day 5, the mice of the CFA, CFA+NE P, CFA+NC P, CFA+4-PSCO NE, and CFA+4-PSCO NC groups showed reductions of 36%, 42%, 44%, 40%, and 33% in paw withdrawal latency, respectively, when compared with the control, NE P, NC P, 4-PSCO NE, and 4-PSCO NC groups. 

As expected, treatment with 4-PSCO free, 4-PSCO NE, and 4-PSCO NC significantly increased paw withdrawal latency in mice exposed to CFA (ANOVA: F _(6, 42)_ = 29.25, *p* < 0.0001), indicating a reversal of thermal sensitivity (Figure 6). The comparison between the groups, per se, can be found in the Appendix A.

### 2.5. CFA Caused Depression-like and Anxiety-like Behaviors in Animals: Therapeutic Potential of 4-PSCO in Free or Nanoencapsulated Form 

The results from the EPM test are presented in Figure 7A,B. The data indicate that CFA-induced mice exhibited a lower percentage of entries into the open arms and spent less time in the open arms compared to the Control, 4-PSCO NE, and 4-PSCO NC groups. Notably, daily treatment with 4-PSCO free, 4-PSCO NE, and 4-PSCO NC reversed the anxiety-like behavior observed in the CFA-induced RA mice, as evidenced by a greater number of entries into the open arms (ANOVA: F _(6, 42)_ = 14.79, *p* < 0.0001). Regarding the time spent in the open arms, the 4-PSCO NE and 4-PSCO NC treatments increased the duration in these arms compared to the CFA group (ANOVA: F _(6, 42)_ = 12.01, *p* < 0.0001). Furthermore, significant differences were observed in the time spent in the open arms between the 4-PSCO NC and 4-PSCO free groups in the EPM test.

Notably, in the TST, the CFA, CFA+NE P, and CFA+NC P groups exposed to CFA showed an increase of 85%, 86%, and 91%, respectively, in immobility time compared to the Control group (ANOVA: F _(6, 42)_ = 2.848, *p* < 0.05). This behavior suggests that the animals in these groups displayed depressive-like behavior. However, none of the treatments were able to reduce depressive behavior in animals exposed to CFA (Figure 7C). The comparison between the groups, per se, can be found in the Appendix A.

### 2.6. Effects of 4-PSCO in Free or Nanoencapsulated Form on AChE and Na^+^, K^+^-ATPase Activities

In the present study, mice that received CFA administration exhibited inhibition of cerebral cortex AChE activity (Figure 8A). Here, 4-PSCO free, 4-PSCO NE, and 4-PSCO NC normalized AChE activity in this tissue after CFA exposure (ANOVA: F _(6, 42)_ = 7.365, *p* < 0.0001) (Figure 8A). However, AChE activity remained unchanged in the spinal cord (Figure 8B) (*p* > 0.05). Similarly, in both tissues, CFA induction or treatment with 4-PSCO (free or its formulations) did not cause any significant changes in Na^+^, K^+^-ATPase enzymatic activity (Figure 8C,D) (*p* > 0.05). The comparison between the groups, per se, is presented in the Appendix A.

### 2.7. Effect of 4-PSCO in Free or Nanoencapsulated Form on Oxidative Stress in CFA-Induced AR and Comorbidities

To confirm the role of oxidative stress in CFA-induced arthritis and related conditions, we measured TBARS levels and CAT activity in the cerebral cortex, spinal cord, and paws of mice (Figure 9). Additionally, TBARS levels were evaluated in the paws. The results showed a significant increase in TBARS levels in the cerebral cortex (~118%) (Figure 9A), spinal cord (~58%) (Figure 9B), and paws (~118%) (Figure 9C) of CFA-induced animals. Treatment with 4-PSCO NC decreased lipid peroxidation caused by CFA in the cerebral cortex (71%) (ANOVA: F _(6, 42)_ = 4.769, *p* < 0.001) and paws (55%) (ANOVA: F _(6, 42)_ = 2.5155, *p* < 0.05). Conversely, 4-PSCO free, 4-PSCO NE, and 4-PSCO NC significantly reduced TBARS levels in the spinal cord (ANOVA: F _(6, 42)_ = 7.100, *p* < 0.0001) (Figure 9B).

Regarding the CAT activity, CFA administration significantly inhibited CAT activity in the cerebral cortex, spinal cord, and paw of mice (Figure 9D–F). Interestingly, treatment with 4-PSCO NE and 4-PSCO NC effectively restored CAT activity in the cerebral cortex (ANOVA: F _(6, 42)_ = 7.518, *p* < 0.0001) (Figure 9D) and paws (ANOVA: F _(6, 42)_ = 9.037, *p* < 0.0001) (Figure 9F), with 4-PSCO NC producing a more pronounced effect. In the spinal cord, both free 4-PSCO and 4-PSCO NC were able to reverse the CFA-induced suppression of CAT activity (ANOVA: F _(6, 42)_ = 3.980, *p* > 0.01) (Figure 9E). Detailed statistical comparisons between groups, per se, are provided in the Appendix A.

## 3. Discussion

In this study, we demonstrate for the first time the pharmacological effects of two different nanostructured systems containing 4-PSCO in mice with RA. Our results reveal differences in the therapeutic effects between the formulations, showing that nanostructured systems can enhance the impact of 4-PSCO. Indeed, both 4-PSCO NC and 4-PSCO NE attenuated the clinical signs and symptoms of CFA-induced arthritis in mice, as well as the inflammatory process. We demonstrated that CFA induction intensified pain sensitivity and increased anxiety and depressive behaviors in the animals, mainly due to oxidative damage in the CNS and peripheral nervous system (PNS). Additionally, CFA induction inhibited AChE activity in the cerebral cortex. Notably, 4-PSCO NC and 4-PSCO NE treatments reversed pain symptoms and anxiety-like behavior in the animals, restored AChE activity, and alleviated oxidative stress. Here, we particularly emphasize the nanoencapsulated form as the most promising. Only the 4-PSCO NC treatment increased the time animals spent in the open arms of the elevated plus-maze test and attenuated CFA-induced oxidative lipid damage, confirming its advantages when compared to free form.

The use of nanostructured platforms such as nanoemulsions and polymeric nanocapsules has significantly advanced drug delivery systems, providing substantial improvements in preserving bioactive compounds and enabling targeted release at the site of action. Compared to other nanocarriers commonly studied in RA, such as liposomes, polymeric micelles, and dendrimers, these systems provide unique benefits. Liposomes, which are made of phospholipid bilayers, can carry both hydrophilic and hydrophobic drugs, but their clinical use is often limited by stability problems, rapid clearance, and high production costs [22,23]. Dendrimers, which are highly branched and uniform macromolecules, allow for precise conjugation and multivalent interactions, showing potential for targeted delivery and immunomodulation; however, concerns remain about cytotoxicity, immunogenicity, and the practicality of large-scale manufacturing [24]. Khan et al. [25] developed a novel strategy for rheumatoid arthritis treatment using a transdermal hydrogel loaded with reverse nanomicelles containing infliximab. The nanocarriers were prepared with the triblock copolymer PCL-PEG-PCL through nanoprecipitation and incorporated into a carbopol hydrogel, with eucalyptus oil serving as a permeation enhancer. The optimized formulation had an average particle size of 72 nm and an encapsulation efficiency of 83%, demonstrating sustained in vitro release and enhanced ex vivo permeation. In a CFA-induced arthritis model, the formulation improved behavioral parameters, biochemical markers, as well as histopathological and radiological findings, underscoring its potential as a transdermal treatment for RA. The polymeric micelles, formed from amphiphilic copolymers, enhance the solubility of poorly water-soluble drugs, although they are susceptible to premature drug release and instability in the bloodstream [26].

In this context, polymeric nanocapsules and nanoemulsions are emerging as promising options. These platforms enhance pharmacological effectiveness by stabilizing unstable molecules, reducing systemic toxicity, and enabling controlled and sustained release, which can ultimately lower the required dosage [24,27,28,29]. The success of these formulations depends on various factors, including the biophysical and biochemical properties of the target molecules and the specifics of the preparation methods [30]. A noteworthy example of these advancements is 4-PSCO, a selenocoumarin molecule, which shows high lipophilicity due to the presence of the selenium atom and the coumarin nucleus in its structure [31,32,33]. These structural features not only enhance gastrointestinal permeability but also aid passage through the blood–brain barrier (BBB) and interaction with plasma proteins, which are essential for targeted drug delivery [31,32,33]. Among different techniques used to prepare nanocapsule suspensions, polymer interfacial deposition stands out because of its accessibility, speed, and low cost. This method, versatile enough for many molecules, such as 4-PSCO, provides a strong platform for encapsulating lipophilic compounds [34].

Despite the progress made and the flexibility of the polymer interfacial deposition method, turning such nanotechnologies into clinically applicable products remains a significant challenge. In this context, new manufacturing techniques, such as microfluidic homogenization, provide promising strategies for achieving scalable and reproducible production while maintaining the integrity of the original method [35]. However, ensuring consistency between batches requires strict control of key parameters, including particle size distribution, encapsulation efficiency, and physicochemical stability. Additionally, it is important to recognize that issues related to sterilization procedures, endotoxin removal, and strict adherence to Good Manufacturing Practices (GMP) are essential steps in moving these nanosystems from experimental research to regulatory approval and clinical use.

Polymeric nanocapsules, often made using polymers like PCL, offer notable benefits in drug delivery. PCL, recognized for its biodegradability, biocompatibility, and mechanical strength, provides a promising option for controlled drug release. Its adaptable nature allows for precise modifications to enhance drug loading, concentration, and release rates, thereby maximizing therapeutic effectiveness and reducing side effects [36,37,38]. The combination of the excellent qualities of 4-PSCO and the advantages of PCL results in high encapsulation efficiency, leading to promising therapeutic outcomes, as shown in this study. The tunability of PCL further enables targeted adjustments to meet specific therapeutic needs, highlighting its potential as a versatile platform for enhanced drug delivery [36,38,39,40].

Nanoemulsions offer another promising avenue for drug delivery, particularly for lipophilic bioactive compounds. Their nanoscale oil droplets and ability to solubilize hydrophobic drugs contribute to improved dissolution rates and enhanced systemic bioavailability [30,41,42]. In this study, the inherent lipophilicity of the molecule supported the stability of the nanoemulsion and polymeric nanocapsule formulations, as evidenced by a low PDI value (<0.2) and macroscopic homogeneity. The maintenance of a negative zeta potential and a slightly acidic pH further ensured the stability of the formulations, highlighting their potential for long-term physical stability and efficient drug delivery [3,42,43]. The diminutive size of the particles or droplets prevents destabilizing phenomena such as sedimentation and coalescence [42,43].

The findings of this study offer valuable and encouraging evidence supporting the therapeutic potential of 4-PSCO nanoformulations in RA. However, some areas require further investigation to strengthen and expand these results. The relatively short duration of the in vivo experiments limits our ability to conclude long-term safety and sustained effectiveness, especially given RA’s chronic nature. Future studies with more extended treatment periods will better simulate the clinical situation. Additionally, while the CFA-induced arthritis model is a well-established and accepted method that accurately reflects many clinical and pathological features of RA, other models, such as collagen-induced arthritis or genetically modified animals, could enhance the generalizability of our findings. The current work used a continuous treatment regimen, which effectively demonstrated both efficacy and safety. However, exploring alternative dosing strategies, such as interval-based regimens, might further improve therapeutic outcomes and clinical relevance.

Considering these results and recognizing the antinociceptive and anti-inflammatory effects, along with the lack of toxicity of the 4-PSCO as shown in previous studies [21], we continued these investigations to thoroughly understand the therapeutic impact of 4-PSCO NE and 4-PSCO NC. We demonstrated that both formulations effectively reduced the signs and symptoms of RA. The experimental arthritis model using CFA reliably reproduces key clinical and pathological features of RA, including joint inflammation, synovial hypertrophy, inflammatory cell infiltration, and pannus formation [30,44,45]. Our findings showed that animals induced with CFA experienced increased paw diameter, circumference, edema, and MPO enzymatic activity. Joint edema is a typical hallmark of RA, and MPO is a specific marker of inflammation, mainly indicating reactive species production and neutrophil activity [46,47]. We also found a significant reduction in these parameters in animals treated with both 4-PSCO NE and 4-PSCO NC, suggesting that 4-PSCO has anti-inflammatory properties capable of reducing immune responses. This is further supported by the decreased neutrophil infiltration, which helps mitigate CFA-induced inflammation.

Moreover, in the context of CFA injection-induced allodynia, mediated by local nociceptor sensitization and neural and immune systemic mechanisms [31,44,45], we observed intense sensitivity in the animals, as evidenced by a decrease in the paw withdrawal threshold. Remarkably, both 4-PSCO NE and 4-PSCO NC attenuated mechanical and thermal sensitivities, proving to be more effective than 4-PSCO free. These results underscore the potential of 4-PSCO nanoformulations as promising candidates for the treatment of RA, offering enhanced efficacy in alleviating inflammation and associated pain symptoms.

Oxidative stress and the inflammatory process are highly intriguing features of RA, which are interconnected. Studies have shown the relationship between oxidative stress and inflammation in RA patients, where both elements coexist [48,49]. In RA, the interaction among T cells, B cells, dendritic cells, macrophages, neutrophils, and fibroblast-like synoviocytes is heavily influenced by lipid metabolites. Changes in lipid metabolism in T cells promote tissue migration, synovial inflammation, and the destruction of cartilage and bone [50,51]. In dendritic cells, lipids not only supply energy and membrane components but also serve as regulatory signals, and chronic inflammation impairs their tolerogenic functions. In macrophages and fibroblast-like synoviocytes, the composition of membrane lipids shapes lipid rafts, thus modulating inflammatory signaling pathways [50]. Free fatty acids, in a dose-dependent manner, increase the secretion of cytokines, chemokines, and matrix-degrading enzymes, partly through activating Toll-like receptor 4 [50]. Lipid rafts, rich in cholesterol and glycosphingolipids, act as platforms for receptor clustering and signal transduction, amplifying pro-inflammatory responses, immune cell infiltration, and endothelial dysfunction, including leukocyte adhesion and vascular permeability [52]. Therefore, recognizing these mechanisms is essential, as they are key factors responsible for initiating and maintaining the pathological processes observed in systemic inflammatory conditions such as RA. In our study, the induction of experimental arthritis by CFA led to a significant rise in lipid peroxidation levels, indicating oxidative damage in both the central and peripheral nervous systems of the animals. In disease states, excessive inflammatory cells in joints can damage cellular structures like lipids, contributing to disease progression. 

The vulnerability of the PNS to reactive species caused by CFA induction arises from the lack of an effective barrier, similar to the BBB, which is present in the CNS. In the CNS, peripheral inflammatory signals can enter through various mechanisms, including direct passage of pro-inflammatory cytokines across the BBB, binding of inflammatory mediators to cytokine receptors on cerebrovascular endothelial cell membranes, and infiltration of immune cells and cytokines from the bloodstream through a compromised BBB [53]. Although the exact mechanism behind the increase in reactive species in the CNS remains unclear, it is reasonable to suggest that crossing the BBB is one possible pathway. This idea is supported by our observation that 4-PSCO NC lowered TBARS levels in the cerebral cortex of the animals, likely helped by the molecule’s lipophilicity, which facilitates its ability to pass through the BBB. In this context, it is plausible that 4-PSCO NC interacts with its receptor complex within lipid rafts, disrupting these microdomains and reducing inflammation and oxidative stress. Consistent with this, we saw that 4-PSCO NC was more effective at reducing these parameters in the cerebral cortex, spinal cord, and paws, which fits with existing research on the efficiency and promising drug effects of this nanostructured system [20,21,54].

In the context of antioxidant activity in RA, it is noteworthy that the complex of antioxidant enzymes is impaired in this condition [55]. Our findings show that CAT activity was significantly decreased in the cerebral cortex, spinal cord, and paw tissues of arthritis mice, consistent with previous studies reporting increased mitochondrial TBARS production and reduced CAT activity in these regions during inflammation [56,57]. Notably, treatment with both 4-PSCO NC and 4-PSCO NE effectively restored CAT activity in the cerebral cortex and paw. Additionally, administration of 4-PSCO free and 4-PSCO NC increased CAT activity in the spinal cord of female mice, suggesting that 4-PSCO may have not only anti-inflammatory but also strong antioxidant effects. These findings indicate that the biochemical changes observed significantly contribute to the therapeutic effects of 4-PSCO in arthritis.

Recent insights from psychiatry suggest a pivotal role of peripheral inflammation, as observed in RA, in the development of mental disorders such as depression and anxiety [8,9]. It is well established that RA patients often exhibit more pronounced symptoms of anxiety and depression, with pain intensity being directly linked to the severity of these mental health issues [13,14]. In our study, animals exposed to CFA demonstrated anxious and depressive-like behaviors, characterized by reduced exploration of open arms and increased immobility. Notably, both 4-PSCO NE and NC reversed these anxious behaviors in the animals. However, none of the treatments managed to reduce the depressive symptoms caused by CFA in the animals.

Furthermore, cholinergic dysfunction preceding the development of RA significantly increases disease incidence and exacerbates both disease symptoms and bone destruction in arthritis mice [58]. Cholinergic activation, manifested through the release of acetylcholine (ACh), exerts its bioactivities via muscarinic or nicotinic ACh receptors [56,57]. Our study observed inhibition of AChE activity in the cerebral cortex of animals with RA. The α7 subunit of nicotinic acetylcholine receptors (α7nAChRs) in macrophages functions to decrease the production of pro-inflammatory cytokines [59]. Thus, it is plausible to suggest that 4-PSCO free, 4-PSCO NE, and 4-PSCO NC can act via the cholinergic anti-inflammatory pathway by binding to α7nAChRs and restoring enzymatic activity, thereby reducing the inflammatory process. This hypothesis aligns with the observed anxiolytic effects, where a reduction in inflammation attenuates these symptoms. However, further investigation into the molecular mechanisms underlying the observed effects is warranted to elucidate the therapeutic potential of 4-PSCO in managing both RA and associated mental health issues.

The promising preclinical results of 4-PSCO NC demonstrate their therapeutic potential; however, several challenges need to be addressed before clinical application. Maintaining consistent particle size, encapsulation efficiency, and physicochemical stability is essential for ensuring reliable efficacy and safety. Additionally, regulatory aspects such as sterilization, endotoxin management, and strict adherence to GMP must be thoroughly addressed. Improving production processes to achieve batch-to-batch consistency and rigorous quality control of excipients (e.g., polymers, medium-chain triglycerides, and surfactants) will be crucial when transitioning from laboratory to clinical-grade products. Despite these hurdles, ongoing progress in nanoparticle manufacturing and the development of standardized protocols strongly indicate that scaling up these nanocapsules for clinical useis achievable, reinforcing their translational potential. Moreover, extending treatment duration, optimizing dosing strategies, and incorporating more preclinical models will be vital to overcoming current limitations and further validating the translational potential of 4-PSCO nanoformulations as safe and effective therapies for RA. Overall, our findings lay a strong foundation for future research and highlight the promise of 4-PSCO and nanotechnology in developing new treatments for RA.

## 4. Materials and Methods

### 4.1. Animals and Ethical Approval

Adult female *Swiss* mice, aged 60 days, were sourced from the Federal University of Pelotas, Brazil. The animals were housed in standard cages under controlled conditions, including a temperature of 22 ± 2 °C and a 12-h light/dark cycle (lights on at 6:00 a.m.), with unrestricted access to food and water. All procedures related to animal care and behavioral testing adhered to the *National Institutes of Health Guide for the Care and Use of Laboratory Animals* (NIH Publications No. 823, revised 1978) and the *International Guiding Principles for Biomedical Research Involving Animals* [60]. The study protocols were reviewed and approved by the Commission on Care and Use of Experimental Resources at the Federal University of Pelotas, affiliated with the National Council for the Control of Animal Experimentation, under protocol number CEEA 13049-2021. Efforts were made to minimize both the number of animals used and any potential discomfort.

### 4.2. Drugs and Reagents

Poly(ℇ-caprolactone) (PCL) (MW: 80 kDa) and sorbitan monooleate (Span^®^ 80) were obtained from Sigma Aldrich (São Paulo, SP, Brazil). The polysorbate 80 (Tween^®^ 80) and medium chain triglycerides (MCT) were provided by Delaware (Porto Alegre, RS, Brazil). Pullulan was kindly donated by Hayashibara (São Paulo, SP, Brazil). Acetone was acquired from Química Moderna (São Paulo, SP, Brazil). High-performance liquid chromatography-grade acetonitrile was obtained from LiChrosolv (São Paulo, SP, Brazil). All other reagents and solvents were of analytical grade and used as received. The compound 4-PSCO depicted in Figure 10 was synthesized and characterized at the Laboratory of Sustainable Chemistry and Metabolomics, Federal University of Santa Maria [61]. The chemical purity of this compound was determined to be 99.9% via GC/MS analysis. 4-PSCO was solubilized in canola oil for administration through the intragastric (i.g.) route at a dosage of 10 mL kg of body weight. The canola oil was obtained from a supermarket (Pelotas, RS, Brazil).

### 4.3. Preparation of Nanocapsule Suspensions and Nanoemulsion Containing 4-PSCO

Three batches of 4-PSCO nanocapsules were produced using the polymer interfacial deposition method [62]. The organic phase, composed of 4-PSCO (10 mg), PCL polymer (100 mg), MCT oil (300 mg), Span^®^ 80 (77 mg), and acetone (27 mL), was kept under moderate magnetic stirring at 40 °C for 2 h. After the complete dissolution of the components, this phase was injected into an aqueous phase (53 mL) containing Tween^®^ 80 (77 mg) and maintained under magnetic stirring at room temperature for 10 min.

The 4-PSCO NE was prepared (3 batches) using the spontaneous emulsification method [32]. An organic phase containing MCT oil (150 mg), Span^®^ 80 (77 mg), and acetone (27 mL) was maintained under moderate magnetic stirring at 40 °C for 45 min. After the complete dissolution of the components, this phase was injected into an aqueous phase (53 mL) containing pullulan (77 mg) and maintained under magnetic stirring at room temperature for 10 min.

For both formulations, the acetone was eliminated under reduced pressure (Rotary evaporator 558, Fisatom^®^), and the formulations were concentrated until the final volume of 10 mL, corresponding to the 4-PSCO concentration of 1 mg/mL. For comparison purposes, placebo formulations (NE P and NC P) were prepared similarly without adding the 4-PSCO into the organic phase. After preparation, the formulations were packaged in glass flasks and characterized in terms of average diameter, polydispersity index (PDI), zeta potential, pH, 4-PSCO content, and encapsulation efficiency (EE%) [62,63].

### 4.4. Analytical Method for the 4-PSCO Quantification

4-PSCO was quantified by the previously validated high-performance liquid chromatography (HPLC) method [21]. The chromatographic system was composed of a UV detector (HPLC-UV) (Shimadzu, Kyoto, Japan, LC-10A system) equipped with an LC-20AT pump, a UV-VIS SPD-M20A detector, a CBM-20A system controller, and a SIL-20A HT autosampler. Column C-18 (Agilent, reversed phase, 5 μm, 110 Å, 150 mm × 4.60 mm) was used. The mobile phase consisted of an isocratic system (30% of ultrapure water and 70% of acetonitrile) at a 1.0 mL/min flow rate. The injection volume was 20 µL, and the amount of 4-PSCO in the samples was determined at 307 nm with a retention time of 5.3 min.

### 4.5. Physicochemical Characterization of the Nanocapsule Suspension and Nanoemulsion

The Average diameter and PDI of the formulations were assessed using dynamic light scattering (DLS) (Zetasizer Nano-ZS ZEN 3600 model, Malvern Instruments, Malvern, UK). Prior to measurement, the samples were diluted 1:500 in ultrapure water that had been filtered through a nylon membrane (0.45 µm, Millipore^®^). Zeta potential was evaluated based on electrophoretic mobility using the same instrument, with the formulations diluted 1:500 in a 10 mM NaCl solution, also pre-filtered (0.45 µm, Millipore^®^). Formulation pH was measured directly using a calibrated potentiometer (Model pH 140, Simpla, Brazil) without prior dilution [62,63].

The total content of 4-PSCO was determined after it was extracted from the formulations. For this, an aliquot of 150 μL of the formulations was placed in a volumetric flask of 10 mL (final concentration of 15 μg mL, corresponding to the midpoint of the standard curve) containing acetonitrile or 30% ultrapure water and 70% acetonitrile for nanocapsule suspensions and nanoemulsions, respectively. Then, a set of ultrasound baths (Ultrasonic bath Q3.0/40A model, Ultronique, Brazil) was applied for 10 min. Following, the samples were filtered (0.45 µm, Millipore^®^) before 4-PSCO content determination by the HPLC method previously described in Section 4.4.

The EE% was determined by ultrafiltration using a 10.000 MW centrifugal device (Amicon^®^ Ultra, Millipore, Darmstadt, Germany). An aliquot of each formulation (300 µL) was placed in the centrifugal device, and centrifugation (30 min at 2200× *g*) was applied. The difference between total and 4-PSCO free concentrations, determined in the formulations and the ultrafiltrate, respectively, was defined as EE% calculated according to Equation (1):


(1)
$$EE\;\%=\frac{Total\;4PSCO\;\;content\;in\;the\;NC\;or\;NE-Free\;4\;PSCO\;\;content}{Total\;4PSCO\;\;content\;in\;the\;NC\;or\;NE}$$


### 4.6. In Vitro Release Profile of 4-PSCO from Polymeric Nanocapsules and Nanoemulsions

The release kinetics of 4-PSCO from both nanoemulsions and polymeric nanocapsules were assessed using the dialysis bag diffusion method (*n* = 3). For this, 1 mL of each formulation was placed into a cellulose dialysis membrane (molecular weight 14,000 Da, Sigma-Aldrich, Saint Louis, MO, USA) and immersed in 200 mL of a release medium composed of 70% 1 M potassium phosphate buffer (pH 7.4) and 30% ethanol. The system was maintained with gentle magnetic stirring at 37 °C. Aliquots of 1 mL from the external medium were collected at predetermined time points (1, 2, 3, 4, 5, 6, 8, 10, 12, 24, and 48 h), and an equal volume of fresh medium was immediately replaced. The concentration of released 4-PSCO was measured using the HPLC method outlined in Section 4.4. Total 4-PSCO content was determined through a standard calibration curve (0.25–10 μg/mL; *r* = 0.99), and the release data were expressed as the percentage of the total compound released.

### 4.7. Experimental Design of In Vivo Evaluations

The mice were randomly divided into twelve groups (*n* = 7 animals/group): (I) Control, (II) NE P, (III) NC P, (IV) 4-PSCO free, (V) 4-PSCO NE, (VI) 4-PSCO NC, (VII) CFA, (VIII) CFA+NE P, (IX) CFA+NC P, (X) CFA+4-PSCO free, (XI) CFA+4-PSCO NE, and (XII) CFA+4-PSCO NC. In the experimental model used in this study, RA was induced by intraplantar (i.pl.) injection of 0.1 mL of Complete Freund’s Adjuvant (CFA) containing 10 mg/mL heat-killed Mycobacterium tuberculosis into the plantar surface of the left hind paw of mice. The animals in groups VII, VIII, IX, X, XI, and XII received an injection of CFA (0.1 mL, i.pl.) in the left hind paw, while the I, II, III, IV, V, and VI groups received 0.9% saline solution (vehicle) in the same volume and administration route on day 0. On day 5 of the experimental protocol, the animals of the Control and CFA groups received canola oil (10 mL kg^−1^, i.g.), NE P and CFA+NE P groups received placebo nanoemulsion (10 mL kg^−1^, i.g.), NC P and CFA+NC P groups received placebo nanocapsule (10 mL kg^−1^, i.g.), 4-PSCO free and CFA+4-PSCO free groups received 4-PSCO (1 mg kg^−1^, i.g.), whereas the 4-PSCO NE and CFA+4PSCO NE groups received 4-PSCO-loaded nanoemulsion (1 mg kg^−1^, i.g.), and 4-PSCO NC and CFA+4-PSCO NC groups received 4-PSCO-loaded polymeric nanocapsule suspension (1 mg kg^−1^, i.g.) until day 15 [64].

Mechanical and thermal sensitivities were assessed using the Von Frey (VF) and Hot-Plate (HP) tests, respectively, on days 0 (baseline), 5, 10, and 15 of the experimental protocol. On day 14, behavioral evaluations addressing locomotor, exploratory, and emotional parameters were performed using the open field test (OFT), elevated plus-maze (EPM), and tail suspension test (TST). Paw inflammation was subsequently quantified on day 15 through measurements of paw edema, diameter, and circumference, in addition to the arthritis scoring. On day 16, animals were euthanized by inhalation of isoflurane, and tissues, including paws, spinal cord, and cerebral cortex, were promptly dissected, weighed, and stored at −20 °C for subsequent biochemical analyses (Figure 11).

### 4.8. Clinical Manifestations of RA

#### 4.8.1. Arthritis Score

On day 15, the clinical severity of polyarthritis was assessed using a visual scoring system ranging from 0 to 4, as described by Zhang et al. [65]. The assessment focused on the presence of edema and erythema in the ipsilateral paw. The scoring criteria were defined as follows: (0) absence of swelling, (1) swelling confined to the toe joints, (2) swelling involving both the toes and toe joints, (3) swelling extending to the ankle joints, and (4) pronounced swelling of the entire paw resulting in impaired mobility.

#### 4.8.2. Assessment of Polyarthritis

The progression of arthritis was recorded on day 15 following CFA induction. Joint diameter (mm) was measured using a Vernier caliper, while paw circumference (cm) was determined by encircling the paw with a string and measuring the length against a ruler [66].

### 4.9. Behavioral Tests

#### 4.9.1. Von Frey Test

As outlined by Alamri et al. [67], the mechanical sensitivity of mice was assessed using a digital analgesimeter (Insight, Ribeirão Preto, SP, Brazil), equipped with a paw pressure transducer. Initially, the animals were allowed to acclimate for 30 min in individual transparent plastic chambers positioned on an elevated wire mesh platform, facilitating access to the plantar surfaces of their paws. Subsequently, the test involved eliciting a hind paw flexion reflex using a digital analgesimeter equipped with a polypropylene tip. The paw withdrawal threshold was measured by a trained investigator who applied the polypropylene tip perpendicularly to the center of the plantar surface of the hind paw, exerting constant progressive pressure until the paw was withdrawn. Data were expressed as withdrawal threshold (g).

#### 4.9.2. Hot-Plate Test

The nociceptive response to thermal stimuli was evaluated as previously documented [68], through the hot-plate test. Each mouse was positioned on a heated metal surface plate (52 ± 1 °C) enclosed by a transparent acrylic box. The latency to display a nociceptive behavior (such as jumping, shaking, or licking the hind paws) was timed from the moment the animal was placed on the apparatus. To prevent any harm to the animal’s paws, a maximum exposure limit of 45 s was set.

#### 4.9.3. Locomotor and Exploratory Domain Assessment

The OFT was employed to evaluate locomotor and exploratory activities in the mice. This test aimed to detect any sedative effects or motor abnormalities that may occur following treatment. The open field apparatus was constructed from plywood and enclosed by walls measuring 30 cm in height. The floor of the open field, measuring 45 cm in length and 45 cm in width, was divided into 9 squares using masking tape markers (arranged in 3 rows of 3). Each mouse was individually placed at the center of the apparatus and observed for a 4-min period to document locomotor activity (measured by the number of squares crossed with all four paws) and exploratory behavior (indicated by the number of times rearing on hind limbs). After each session, the arena was cleaned with a 20–30% ethanol solution, and each mouse was tested only once [69].

### 4.10. Evaluation of Emotional Responses

#### 4.10.1. Assessment of Anxiety-Related Behavior

Following the methodology of Pellow et al. [70], the EPM test was performed on day 14 to evaluate anxiety-like behavior in rodents. The EPM apparatus consisted of a cross-shaped platform with two open arms (16 cm × 5 cm) and two closed arms (16 cm × 5 cm × 10 cm), converging at a central square (5 cm × 5 cm) positioned 50 cm above the floor. Each rodent was individually placed on the central platform facing an open arm. During a 5-min observation period, the number of entries and the amount of time spent in both open and closed arms were recorded. An anxiolytic effect is inferred from a significant increase in exploration of the open arms.

#### 4.10.2. Assessment of Depression-Related Behavior

The TST was conducted following the methodology outlined by Steru et al. [71]. Mice were suspended 50 cm above the ground using adhesive tape placed approximately 1 cm from the tip of their tails. Immobility in mice was defined as a state of complete and passive inactivity. An observer with extensive experience manually timed the duration of immobility over a 6-min period. A reduction in immobility time is typically interpreted as an indication of an antidepressant-like effect.

### 4.11. Ex Vivo Assays

#### 4.11.1. Tissue Processing for Biochemical Analyses

Paw tissues were collected to assess edema and MPO activity. For MPO determination, the paw samples were finely chopped, pooled, and homogenized in phosphate-buffered saline (PBS; 20 mmol L, pH 7.4) containing 0.1 mmol/L ethylenediaminetetraacetic acid. The homogenates were first centrifuged at 2000 rpm for 10 min at 4 °C to obtain the supernatant fraction (S_1_). The S_1_ fraction was then subjected to a second centrifugation at 14,000 rpm for 15 min at 4 °C, yielding a final pellet (P_2_). This pellet was resuspended in a solution containing 50 mmol L potassium phosphate buffer (pH 6.0) and 0.5% (*w*/*v*) hexadecyltrimethylammonium bromide. Samples underwent three cycles of freezing and thawing before enzymatic analysis.

Samples of the cerebral cortex and spinal cord were homogenized in cold 50 mM Tris-HCl at pH 7.4 (1/5 weight/volume) for biochemical analyses, including thiobarbituric acid reactive substances (TBARS), catalase (CAT), and Na^+^, K^+^-ATPase. The homogenates were centrifuged at 3000 rpm for 10 min at 4 °C to obtain the S_1_. For the determination of nitrite and nitrate (NOx) levels, tissues were homogenized in 200 mM ZnSO_4_ and 96% acetonitrile, followed by centrifugation at 14,000 rpm for 30 min at 4 °C. The resulting supernatant was then used for the NOx assay. AChE activity was determined by preparing the samples in 0.25 mol/L sucrose buffer (1/10 weight/volume) and centrifuging at 3000 rpm for 10 min to produce the S_1_ fraction.

The protein content in the S_1_ fraction was quantified using the Bradford assay [72], with bovine serum albumin (1 mg mL^−1^) serving as the standard. Absorbance was measured at 595 nm using a spectrophotometer.

#### 4.11.2. Inflammatory Parameters

##### Paw Edema

On day 15 of the experimental protocol, paw edema was evaluated. Mice were euthanized, and both paws were harvested. The edema was determined by calculating the weight difference between the control (left) and CFA-injected (right) paws using an analytical balance, with values expressed in grams (g).

##### MPO Assay

MPO activity was measured using a modified version of a previously described method [73]. In this assay, a 10 μL aliquot of resuspended P_2_ was added to a medium containing resuspension medium and N,N,N′, N′-tetramethylbenzidine (1.5 mM). The kinetic analysis of MPO began with the addition of 3.5% hydrogen peroxide (H_2_O_2_) (0.01% *v*/*v*), and the resulting color change was measured using a Molecular Devices SPECTRAmax Plus 384 spectrophotometer at 655 nm and 37 °C. The results were expressed as optical density/milligrams protein/minute.

#### 4.11.3. Oxidative Stress Parameters

##### TBARS Levels

TBARS levels, a measure of lipid peroxidation, were determined as described by Ohkawa et al. [74]. An aliquot (40 μL) of S_1_ was incubated with 0.8% thiobarbituric acid (100 μL), acetic acid buffer pH 3.4 (100 μL), and 8.1% sodium dodecyl sulfate (40 μL) at 95 °C for 2 h. The color reaction was measured at 532 nm (spectrophotometer Molecular Devices SPECTRAmax Plus 384). TBARS levels were expressed as nmol MDA/milligrams protein.

##### CAT Activity

CAT activity was spectrophotometrically determined by the method of Aebi [75], which involves monitoring the disappearance of hydrogen peroxide (H_2_O_2_) in the presence of the homogenate at 240 nm. The enzymatic assay was started by combining an aliquot of the S1 fraction with the substrate (H_2_O_2_) to achieve a final concentration of 0.3 mM in 50 mM potassium phosphate buffer (pH 7.0). Catalase activity was quantified as U CAT/milligram of protein, where 1 U represents the decomposition of 1 μmol of H_2_O_2_ per minute at pH 7.0 and 25 °C.

#### 4.11.4. AChE Activity

The AChE activity was measured using a modified version of the method described by Ellman [76]. The reaction mixture consisted of 10 μL of S_1_, 100 mM potassium phosphate buffer at pH 7.5 (180 μL), and 1 mM DTNB (20 μL). This method relies on the formation of the yellow anion, 5,5′-dithiobis-nitrobenzoate, which is quantified spectrophotometrically at 412 nm over a 2-min period. The enzyme was pre-incubated for 2 min at 25 °C. The reaction commenced with the addition of 0.8 mM acetylthiocholine iodide (5 μL), and the enzymatic activity was quantified in terms of μmol/h/milligrams protein.

#### 4.11.5. Na^+^, K^+^-ATPase Activity

Samples of the cerebral cortex and spinal cord (S_1_) (12.5 μL) were preincubated at 37 °C for 10 min with a reaction mixture containing 3 mM MgCl, 125 mM NaCl, 20 mM KCl, and 50 mM Tris/HCl, pH 7.4, in a final volume of 229.5 μL. Control samples were carried out under the same conditions with the addition of 0.1 mM ouabain, an inhibitor of the Na^+^, K^+^ pump. The reactions were initiated by adding 3 mM ATP (12.5 μL), followed by incubation of the samples at 37 °C for 30 min. Incubation was then stopped by adding TCA (10%) with 10 mM HgCl_2_ (62.5 μL). Na^+^, K^+^-ATPase activity was calculated through the difference between the two assays. Released inorganic phosphate (Pi) was measured using the method of Fiske and Subbarow [77], and the enzyme activity was expressed as nmol Pi/milligrams protein/minute.

### 4.12. Statistical Analysis

All experimental findings are presented as the mean ± standard error of the mean (SEM). Statistical analysis was performed using the GraphPad Prism 6.0 and 8.0 software (Graph Pad, San Diego, CA, USA). The normality of the data was evaluated using the D’Agostino and Pearson omnibus and the Shapiro–Wilk normality test. Data were analyzed by one-way ANOVA, followed by Tukey’s test or Unpaired Student’s *t*-test when appropriate. Probability values below 0.05 (*p* < 0.05) were considered statistically significant.

## 5. Conclusions

In summary, our results show that both 4-PSCO NE and 4-PSCO NC significantly reduced clinical symptoms, inflammation signs, and nociceptive responses in the CFA-induced arthritis model in female mice. These formulations also demonstrated novel anxiolytic effects in this model, which seem to be linked to the reduction in inflammation and oxidative stress in both the CNS and PNS, as well as the modulation of AChE activity in the cerebral cortex. These mechanisms together help relieve pain and prevent neuropsychiatric comorbidities.

Notably, only the 4-PSCO NC demonstrated superior efficacy in increasing the time spent by animals in the open arms in the EPM test compared to the free 4-PSCO. Additionally, it significantly reduced TBARS levels in the cerebral cortex and paw, as well as MPO enzymatic activity in the paw. These findings suggest that nanoencapsulation enhances the therapeutic profile of 4-PSCO. The high lipophilicity of the molecule, combined with the physicochemical properties of the nanoformulation, may facilitate its permeation across the BBB, potentially leading to improved efficacy, safety, specificity, and neuroprotective potential in disease management. These findings underscore the role of nanotechnology as an innovative and effective platform for developing therapies for chronic inflammatory diseases such as RA.

## Figures and Tables

**Figure 1 pharmaceuticals-18-01379-f001:**
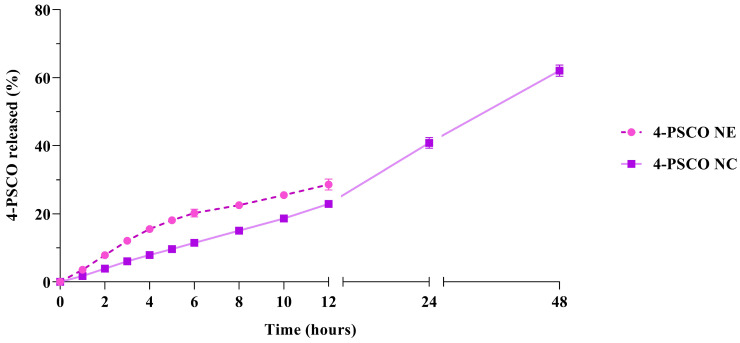
In vitro release profiles of 4-PSCO from nanoemulsion (NE, pink line) and nanocapsules (NC, purple line). The values are reported as mean ± SD of three independent experiments/group.

**Figure 2 pharmaceuticals-18-01379-f002:**
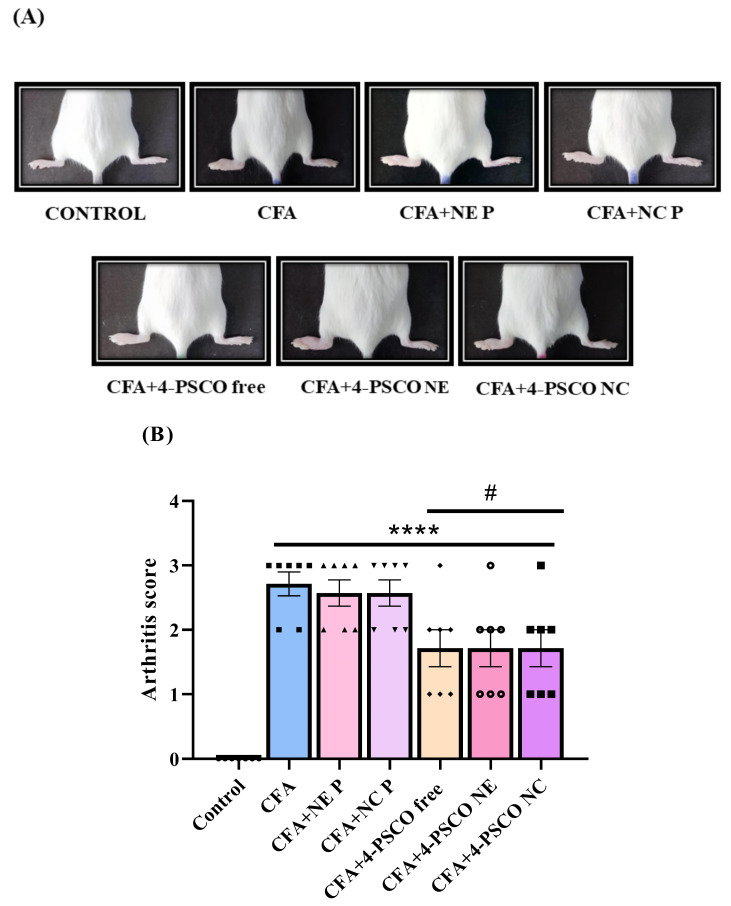
Effects of free 4-(phenylselanyl)-2H-chromen-2-one (4-PSCO), 4-PSCO-loaded nanoemulsions (4-PSCO NE), and 4-PSCO-loaded nanocapsules (4-PSCO NC) (1 mg kg^−1^, i.g.) on rheumatoid arthritis (RA)-like symptoms induced by Complete Freund’s Adjuvant (CFA) (0.1 mL, i.pl.) in mice. Hind paw images (**A**) and the average arthritis score (**B**). Each point represents the mean of 7 female mice per group. (****) *p* < 0.0001 denotes significance compared to the Control group; (#) *p* < 0.05 denotes significance compared to the CFA group (one-way ANOVA followed by Tukey’s test).

**Figure 3 pharmaceuticals-18-01379-f003:**
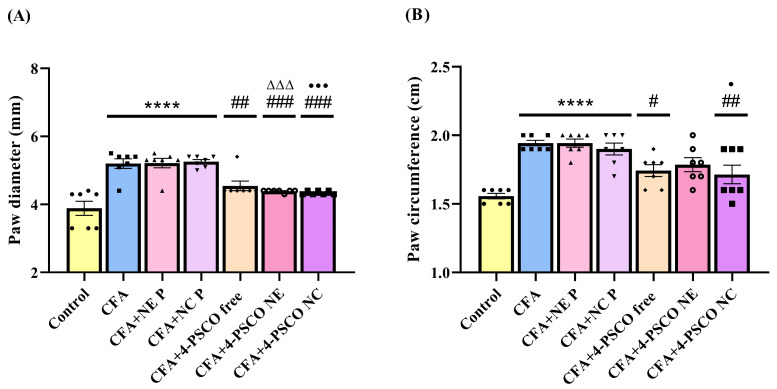
Effects of free 4-(phenylselanyl)-2H-chromen-2-one (4-PSCO), 4-PSCO-loaded nanoemulsions (4-PSCO NE), and 4-PSCO-loaded nanocapsules (4-PSCO NC) (1 mg kg^−1^, i.g.) on paw diameter (**A**) and circumference (**B**) induced by Complete Freund’s Adjuvant (CFA) (0.1 mL, i.pl.) in mice. Each point represents the mean of 7 female mice per group. (****) *p* < 0.0001 denotes significance compared with the Control group; (#) *p* < 0.05, (##) *p* < 0.01, and (###) *p* < 0.001 denote significance compared with the CFA group; (ΔΔΔ) *p* < 0.001 denotes significance compared with the CFA+NE P group; (•) *p* < 0.05, and (•••) *p* < 0.001 denote significance compared with the CFA+NC P group (one-way ANOVA followed by Tukey’s test).

**Figure 4 pharmaceuticals-18-01379-f004:**
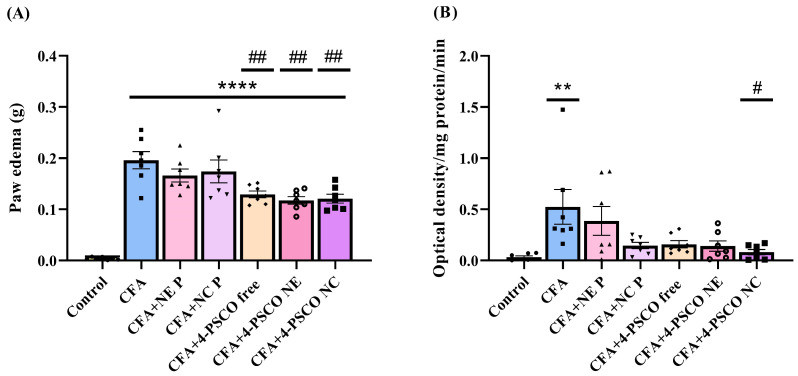
Effects of free 4-(phenylselanyl)-2H-chromen-2-one (4-PSCO), 4-PSCO-loaded nanoemulsions (4-PSCO NE), and 4-PSCO-loaded nanocapsules (4-PSCO NC) (1 mg kg^−1^, i.g.) on paw edema (**A**) and myeloperoxidase (MPO) activity (**B**) induced by Complete Freund’s Adjuvant (CFA) (0.1 mL, i.pl.) in mice. Each point shows the mean of 7 female mice per group. (**) *p* < 0.001 and (****) *p* < 0.0001 indicate significance compared with the Control group; (#) *p* < 0.05 and (##) *p* < 0.01 indicate significance compared with the CFA group (one-way ANOVA followed by Tukey’s test).

**Figure 5 pharmaceuticals-18-01379-f005:**
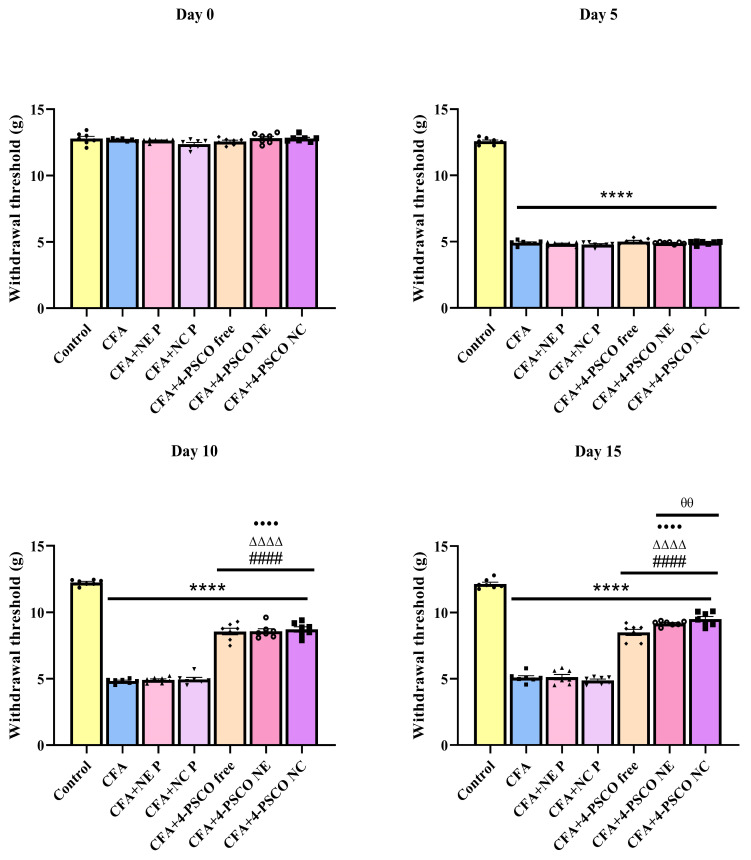
Effects of free 4-(phenylselanyl)-2H-chromen-2-one (4-PSCO), 4-PSCO-loaded nanoemulsions (4-PSCO NE), and 4-PSCO-loaded nanocapsules (4-PSCO NC) (1 mg kg^−1^, i.g.) on paw withdrawal threshold for mechanical stimuli in the von Frey test induced by Complete Freund’s Adjuvant (CFA) (0.1 mL, i.pl.) in mice. Each point represents the mean of 7 female mice per group. (****) *p* < 0.0001 indicates significance compared to the Control group; (####) *p* < 0.0001 indicates significance compared to the CFA group; (ΔΔΔΔ) *p* < 0.0001 indicates significance compared to the CFA+NE P group; (••••) *p* < 0.0001 indicates significance compared to the CFA+NC P group; (θθ) *p* < 0.01 indicates significance compared to the CFA+4-PSCO free group (one-way ANOVA followed by Tukey’s test).

**Figure 6 pharmaceuticals-18-01379-f006:**
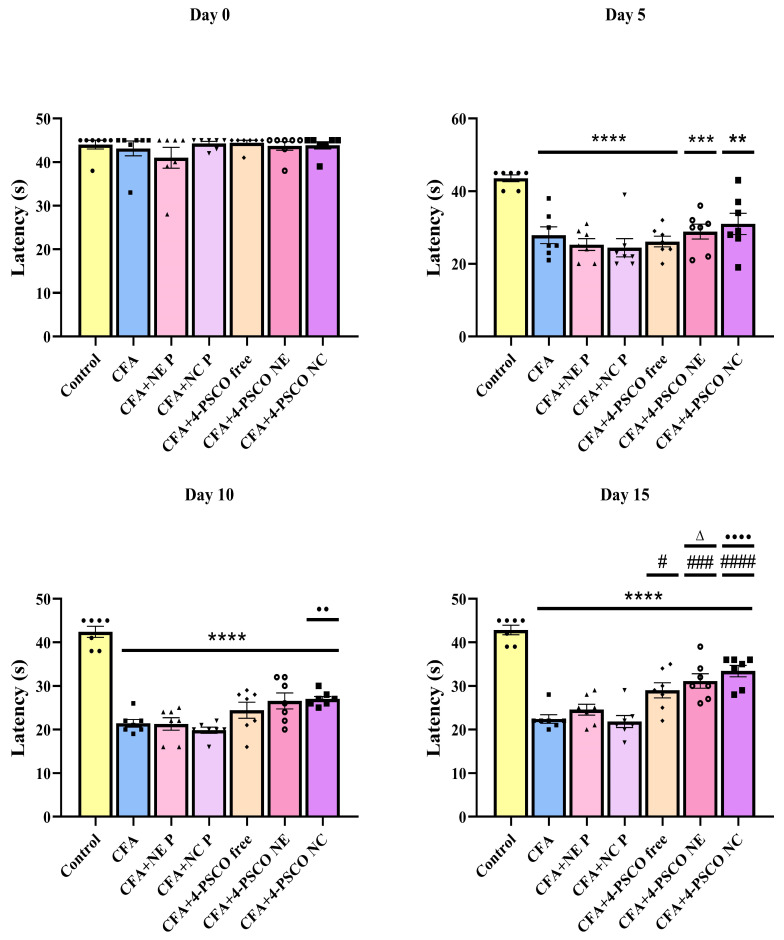
Effects of free 4-(phenylselanyl)-2H-chromen-2-one (4-PSCO), 4-PSCO-loaded nanoemulsions (4-PSCO NE), and 4-PSCO-loaded nanocapsules (4-PSCO NC) (1 mg kg^−1^, i.g.) on thermal hyperalgesia in the hot-plate test induced by Complete Freund’s Adjuvant (CFA) (0.1 mL, i.pl.) in mice. Each point represents the mean of 7 female mice per group. (**) *p* < 0.01, (***) *p* < 0.001, and (****) *p* < 0.0001 denotes significance levels compared to the Control group; (#) *p* < 0.05, (###) *p* < 0.001, and (####) *p* < 0.0001 denote significance compared to the CFA group; (Δ) *p* < 0.05 denotes significance compared to the CFA+NE B group; (••) *p* < 0.01, and (••••) *p* < 0.0001 denote significance compared to the CFA+NC P group (one-way ANOVA followed by Tukey’s test).

**Figure 7 pharmaceuticals-18-01379-f007:**
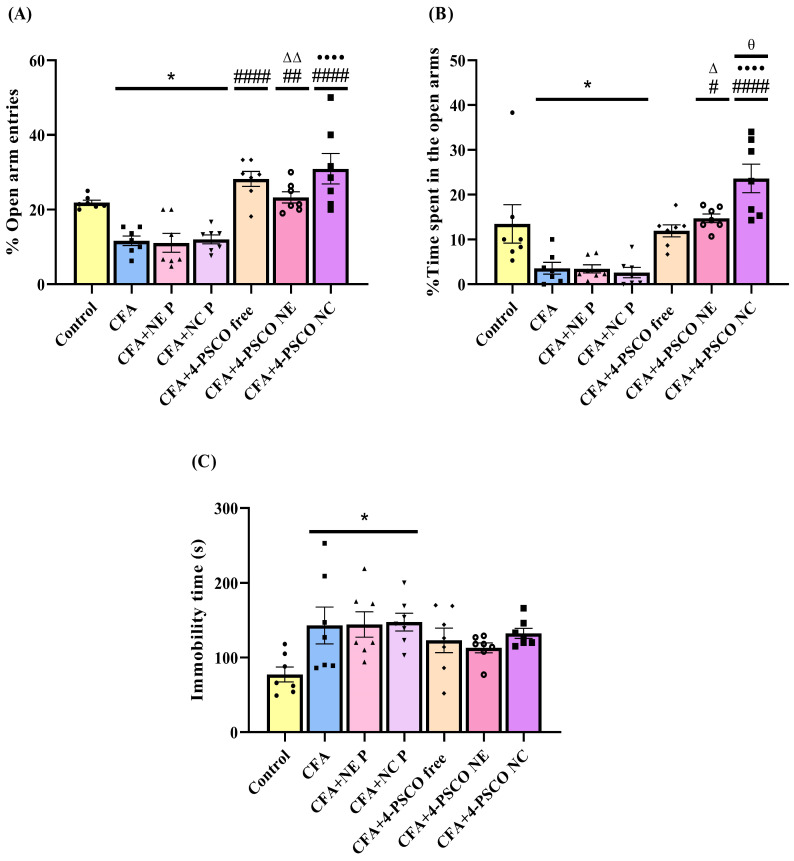
Effects of free 4-(phenylselanyl)-2H-chromen-2-one (4-PSCO), 4-PSCO-loaded nanoemulsions (4-PSCO NE), and 4-PSCO-loaded nanocapsules (4-PSCO NC) (1 mg/kg, i.g.), and Complete Freund’s Adjuvant (CFA) (0.1 mL, i.pl.), on the percentage of entries in the open arms (**A**), percentage of time spent in the open arms (**B**), and immobility time (**C**) in mice. Each point represents the mean of 7 female mice in each group. (*) *p* < 0.05 denotes significance compared to the Control group; (#) *p* < 0.05, (##) *p* < 0.01, and (####) *p* < 0.0001 denote significance compared to the CFA group; (Δ) *p* < 0.05, and (ΔΔ) *p* < 0.01 denote significance compared to the CFA+NE B group; (••••) *p* < 0.0001 denote significance compared to the CFA+NC P group; (θ) *p* < 0.05 denotes significance compared to the CFA+4-PSCO free group. One-way ANOVA followed by Tukey’s test was used for analysis.

**Figure 8 pharmaceuticals-18-01379-f008:**
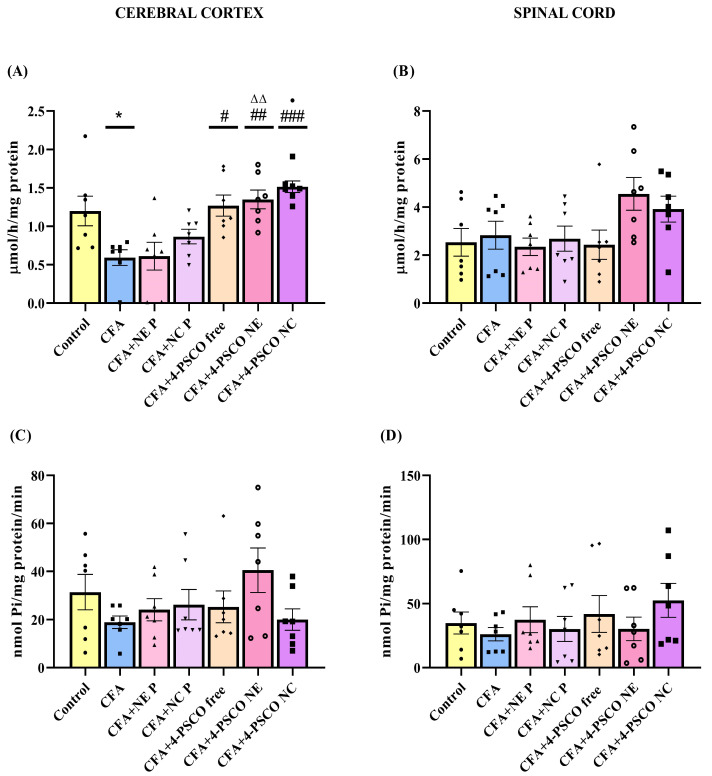
Effects of free 4-(phenylselanyl)-2H-chromen-2-one (4-PSCO), 4-PSCO-loaded nanoemulsions (4-PSCO NE), and 4-PSCO-loaded nanocapsules (4-PSCO NC) (1 mg/kg, i.g.) on AChE activity in the cerebral cortex (**A**) and spinal cord (**B**), as well as Na^+^, K^+^-ATPase activity in the cerebral cortex (**C**) and spinal cord (**D**), induced by Complete Freund’s Adjuvant (CFA) (0.1 mL, i.pl.) in mice. Each point represents the mean of 7 female mice in each group. (*) *p* < 0.05 denotes significance compared with the Control group; (#) *p* < 0.05, (##) *p* < 0.01, and (###) *p* < 0.001 denote significance compared with the CFA group; (ΔΔ) *p* < 0.01 denotes significance compared with the CFA+NE P group; (•) *p* < 0.05 denotes significance compared with the CFA+NC P group (one-way ANOVA followed by Tukey’s test).

**Figure 9 pharmaceuticals-18-01379-f009:**
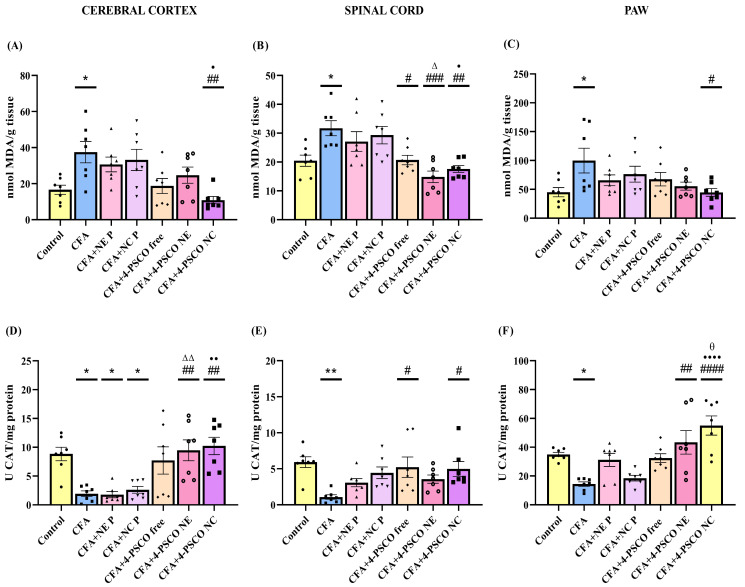
Effects of free 4-(phenylselanyl)-2H-chromen-2-one (4-PSCO), 4-PSCO-loaded nanoemulsions (4-PSCO NE), and 4-PSCO-loaded nanocapsules (4-PSCO NC) (1 mg/kg, i.g.) on TBARS levels in the cerebral cortex (**A**), spinal cord (**B**), and paw (**C**); and CAT activity in the cerebral cortex (**D**), spinal cord (**E**), and paw **(F**) induced by Complete Freund’s Adjuvant (CFA) (0.1 mL, i.pl.) in mice. Each point represents the mean of 7 female mice per group. (*) *p* < 0.05, and (**) *p* < 0.01 denote significance compared to the Control group; (#) *p* < 0.05, (##) *p* < 0.01, (###) *p* < 0.001, and (####) *p* < 0.0001 denote significance compared to the CFA group; (Δ) *p* < 0.05, and (ΔΔ) *p* < 0.01 denote significance compared to the CFA+NE B group; (•) *p* < 0.05, (••) *p* < 0.01, and (••••) *p* < 0.0001 denote significance compared to the CFA+NC P group; (θ) *p* < 0.05 denotes significance compared to the CFA+4-PSCO free group. One-way ANOVA followed by Tukey’s test.

**Figure 10 pharmaceuticals-18-01379-f010:**
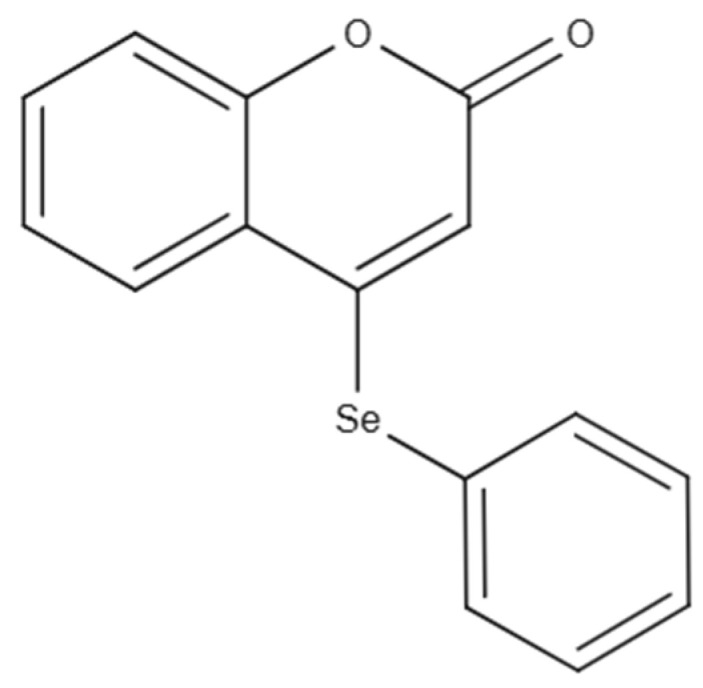
Chemical structure of 4-PSCO.

**Figure 11 pharmaceuticals-18-01379-f011:**
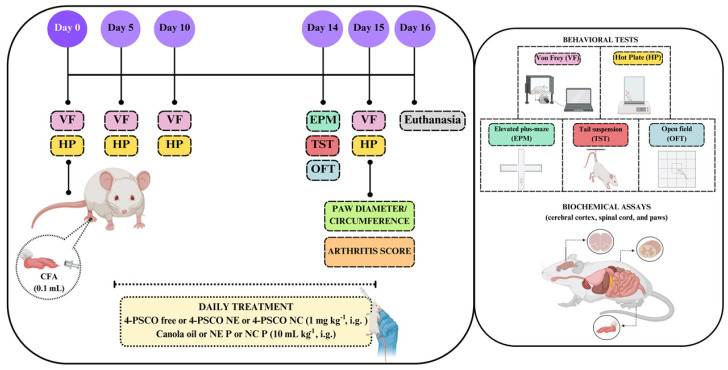
Schematic representation of the experimental design of the study. VF: Von Frey; HP: Hot-plate; EPM: elevated plus-maze; TST: tail suspension test; OFT: open field test.

**Table 1 pharmaceuticals-18-01379-t001:** Physicochemical parameters of the nanocapsules and nanoemulsions containing the 4-PSCO compound and their respective placebo formulations (placebo nanoemulsions (NE P) and placebo nanocapsule (NC P)).

	Parameters				
	Average Diameter (nm)	PDI ^a^	ZP ^b^ (mV)	pH	4-PSCO content (%)
	Nanocapsules (NC)				
4-PSCO NC	251 ± 13	0.19 ± 0.05	−11.4 ± 0.2	5.4 ± 0.0	101.9 ± 1.8
NC P	234 ± 3	0.16 ± 0.01	−11.0 ± 1.5	5.3 ± 0.1	- ^c^
	Nanoemulsions (NE)				
4-PSCO NE	205 ± 2	0.10 ± 0.01	−24.3 ± 0.3	5.4 ± 0.0	100.5 ± 0.7
NE P	208 ± 5	0.13 ± 0.02	−32.01 ± 2 **	5.4 ± 0.1	- ^c^

The results are presented as the mean ± standard deviation (SD) from three independent batches per formulation. Statistical comparisons were performed using an unpaired Student’s *t*-test (** *p* < 0.01). PDI ^a^: Polydispersity index; ZP ^b^: Zeta potential; - ^c^: Not applicable.

## Data Availability

The data will be made available upon request.

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
