# Peer review of "Preclinical Evaluation of Nanoemulsion and Polymeric Nanocapsule Delivery Systems of 4-(Phenylselanyl)-2H-Chromen-2-One for Rheumatoid Arthritis and Comorbidities"

_pharmaceuticals, 2025, doi:10.3390/ph18091379_

Round 1
Reviewer 1 Report
Comments and Suggestions for Authors
This work is the repetition of the same author's work published in https://doi.org/10.1021/acsomega.5c02655 https://doi.org/10.3390/pharmaceutics16020269 The only difference is that they keep changing the delivery system etc. Further, after a careful comparison, it is my opinion that the method proposed didn't provide any greater benefits than the other strategies already reported in the literature. Overall, I don't see any significance in this work that needs attention or that deserve publication in this journal.
Author Response
REVIEWER 1
We would like to thank the reviewer for their suggestions and corrections to the manuscript. We recognize that your comments significantly contribute to improving the quality of the manuscript. The responses to the questions raised are given in sequence.
Reviewer comment:
- Comment - This work is the repetition of the same author's work published in https://doi.org/10.1021/acsomega.5c02655, https://doi.org/10.3390/pharmaceutics16020269. The only difference is that they keep changing the delivery system etc. Further, after a careful comparison, it is my opinion that the method proposed didn't provide any greater benefits than the other strategies already reported in the literature. Overall, I don't see any significance in this work that needs attention or that deserve publication in this journal.
Answer: We thank the reviewer for their careful reading and valuable comments. However, we respectfully disagree with the observation, as the studies cited are fundamentally different from the scope and objectives of the present work. The preceding study of our research group (https://doi.org/10.3390/pharmaceutics16020269) investigated the potential interactions of 4-phenylselanyl-2H-chromen-2-one (4-PSCO) with pain-related proteins through molecular docking analysis. Subsequently, nanocapsule suspensions containing 4-PSCO were developed, using ethylcellulose as the polymeric membrane and medium-chain triglycerides as the oil core. The preliminary toxicity of both free and nanoencapsulated 4-PSCO was evaluated in Caenorhabditis elegans (C. elegans) and Swiss mice. Finally, acute inflammation models were employed to evaluate the pharmacological effects of 4-PSCO in both its free and nanoencapsulated forms in male and female mice.
Regarding the results, 4-PSCO displayed high affinity for pain- and inflammation-related receptors, including p38 MAP kinase, peptidyl arginine deiminase type 4, phosphoinositide 3-kinase, Janus kinase 2, toll-like receptor 4, and nuclear factor-kappa β. The 4-PSCO-loaded nanocapsule suspensions promoted controlled release of the compound and showed no toxicity in C. elegans and mice. Both free and nanoencapsulated forms demonstrated antinociceptive and anti-inflammatory effects in acute inflammatory pain models.
In the second study mentioned (https://doi.org/10.1021/acsomega.5c02655), we explored the therapeutic effects of free 4-PSCO on inflammatory and oxidative parameters during aging. It is well established that both the structure and function of the organism decline with age, and aging is a major risk factor for several chronic diseases, including rheumatoid arthritis (RA). In the context of RA, aging plays a crucial role in disease progression and severity. Our findings demonstrated that treatment with 4-PSCO alleviated pain sensitivity, reduced inflammation, and restored both body weight and spleen index in aged animals. Moreover, 4-PSCO attenuated oxidative stress in the paw and spinal cord.
Importantly, unlike the current and previous studies, no nanostructured formulation was investigated, and sex-related differences were not addressed, primarily due to the limited availability of aged animals, financial constraints, and adherence to the ethical principles of the 3Rs (Replacement, Reduction, and Refinement). Therefore, the objectives, scope, and focus of that study differ substantially from both the present work and our previous nanocarrier-based approaches.
By contrast, the current manuscript proposes the development of distinct nanostructured systems, including a polymeric nanocapsule made from Poly(Ε-Caprolactone) polymer and a nanoemulsion. The rationale was to optimize and further enhance the pharmacological performance of 4-PSCO by exploring alternative nanosystems. In our previous study (https://doi.org/10.3390/pharmaceutics16020269), the nanocapsule suspension was developed using ethylcellulose, a hydrophobic cellulose derivative recognized for its high controllability in drug release [1]. Although nanoencapsulated 4-PSCO demonstrated superior pharmacological performance compared to the free form in certain parameters, we hypothesized that changes in the qualitative composition of the formulation could further enhance the compound's pharmacological performance.
Therefore, while our earlier findings were indeed promising, the present work provides a clear advancement by (i) developing different nanocarrier systems, (ii) evaluating a relevant chronic disease model, and (iii) aiming to broaden the therapeutic applicability of 4-PSCO. We believe this demonstrates both the originality and scientific contribution of our study, reinforcing that it is not a repetition of prior work, but rather a logical and ethical continuation toward the discovery of new therapeutic strategies for chronic inflammatory conditions such as rheumatoid arthritis and its associated comorbidities.
References:
- Adeleke OA. Premium ethylcellulose polymer based architectures at work in drug delivery. International Journal of Pharmaceutics: X 2019, 8, 1, 100023. https://doi: 10.1016/j.ijpx.2019.100023

Reviewer 2 Report
Comments and Suggestions for Authors
1. The introduction describes the complex and multifactorial nature of RA but does not include epidemiological statistics. Could the authors provide up-to-date global or regional prevalence and incidence data for RA to better emphasize the public health burden and the importance of developing new therapies?
2. The manuscript highlights nanotechnology as an emerging approach for RA but does not compare 4-PSCO nanocarrier formulations with other nanotechnology-based RA treatments reported in the literature. Could the authors briefly discuss existing nanocarrier systems (e.g., liposomes, micelles, dendrimers) and how their approach potentially improves upon these?
3. While the introduction notes the relationship between RA, oxidative stress, and psychiatric comorbidities (anxiety, depression), the cited references ([8–12]) appear broad. Could the authors include more RA-specific and recent studies (within the last five years) that directly investigate these associations to strengthen the background?
4. The authors mention prior molecular docking results showing high binding affinity of 4-PSCO to pain- and inflammation-related proteins but do not specify the exact targets. Could these be explicitly named (e.g., COX-2, TNF-α, IL-1β receptors) in the introduction to clarify the mechanistic rationale?
5. The choice to evaluate both nanoemulsion and polymeric nanocapsule systems is stated, but the reasoning is not fully clear. Could the authors elaborate on why both formulations were pursued—was this for comparative performance assessment, differences in drug release kinetics, stability, or targeted tissue penetration?
6. While the introduction gives a general overview of RA pathophysiology and nanotechnology potential, the specific knowledge gap this work addresses could be more explicitly stated. For example, the authors could emphasize the unmet need for RA treatments that simultaneously target joint inflammation, modulate CNS oxidative stress, and address psychiatric comorbidities. Clarifying how current therapeutic and nanocarrier approaches fail to address these aspects would highlight the novelty of their study.
7. The current transition from RA pathophysiology to nanotechnology applications is somewhat abrupt. Adding a bridging paragraph that connects RA’s biochemical features (e.g., oxidative stress, BBB permeability issues, immune dysregulation) to the rationale for using a lipophilic compound like 4-PSCO and encapsulating it in nanocarriers could improve readability. This would also strengthen the scientific justification for why these formulations may outperform standard delivery methods.
Author Response
REVIEWER 2
We would like to thank the reviewer for their suggestions and corrections to the manuscript. We recognize that your comments contribute to the improvement of the manuscript's quality. The responses to the questions raised are given in sequence.
Reviewer comments:
- Comment - The introduction describes the complex and multifactorial nature of RA but does not include epidemiological statistics. Could the authors provide up-to-date global or regional prevalence and incidence data for RA to better emphasize the public health burden and the importance of developing new therapies?
Answer: We thank the reviewer for the valuable comment. In response, we have revised the Introduction section to include a more detailed overview of the latest prevalence and incidence data on RA, highlighting both its public health impact and the need for new therapeutic approaches (Introduction, Page 2).
- Comment - The manuscript highlights nanotechnology as an emerging approach for RA but does not compare 4-PSCO nanocarrier formulations with other nanotechnology-based RA treatments reported in the literature. Could the authors briefly discuss existing nanocarrier systems (e.g., liposomes, micelles, dendrimers) and how their approach potentially improves upon these?
Answer: Ok. We thank the reviewer for their comment. As suggested, a brief discussion of existing nanocarrier systems was included in the manuscript (Discussion section, page 14).
- Comment - While the introduction notes the relationship between RA, oxidative stress, and psychiatric comorbidities (anxiety, depression), the cited references ([8–12]) appear broad. Could the authors include more RA-specific and recent studies (within the last five years) that directly investigate these associations to strengthen the background?
Answer: We thank the reviewer for their valuable feedback. However, after conducting an additional search, we found very few studies directly correlating oxidative stress with psychiatric disorders in the context of rheumatoid arthritis. For this reason, we decided to retain the references already cited in our manuscript. Moreover, this further reinforces the novelty of our study, as it comprehensively addresses these interrelated aspects (Introduction session, page 2).
- Comment - The authors mention prior molecular docking results showing high binding affinity of 4-PSCO to pain- and inflammation-related proteins but do not specify the exact targets. Could these be explicitly named (e.g., COX-2, TNF-α, IL-1β receptors) in the introduction to clarify the mechanistic rationale?
Answer: We thank the reviewer for the relevant comment. Yes, we can certainly add the proteins related to pain and inflammation that were evaluated in our previous study to the Introduction section. We would like to note, however, that these proteins were not initially included to keep the Introduction more concise and focused. Nevertheless, we agree that their inclusion can further strengthen the context, and we will incorporate them as suggested (Introduction section, page 2).
- Comment - The choice to evaluate both nanoemulsion and polymeric nanocapsule systems is stated, but the reasoning is not fully clear. Could the authors elaborate on why both formulations were pursued—was this for comparative performance assessment, differences in drug release kinetics, stability, or targeted tissue penetration?
Answer: We thank the reviewer for bringing this important point to our attention. The decision to investigate both nanoemulsion and polymeric nanocapsule systems was deliberate and grounded in their complementary characteristics, which we considered relevant for optimizing the administration of 4-PSCO. Nanoemulsions offer a relatively simple preparation process, high solubilization capacity for lipophilic compounds, and the potential for rapid drug release, thereby favoring an early pharmacological effect. In contrast, polymeric nanocapsules provide a more controlled and sustained release profile.
In this context, we evaluated both systems in parallel to compare their performance and release profiles and identify the most promising platform for enhancing the therapeutic profile of 4-PSCO. It is worth noting that the formulations differ substantially in composition: nanocapsules are composed of a polymeric membrane, while nanoemulsions consist of oil droplets. The polymer poly(ε-caprolactone) (PCL) was selected due to its biocompatible and biodegradable properties, as well as its controlled release properties [1-3].
Nanoemulsions, on the other hand, are colloidal dispersions of immiscible liquids stabilized by surfactants and are widely recognized for improving drug solubility, bioavailability, and dissolution rate, encompassing hydrophobic, hydrophilic, and amphiphilic molecules [5-6]. Unlike polymeric nanocapsules, nanoemulsions lack a polymeric membrane, which may reduce risks associated with toxicity or unwanted interactions with living cells. Therefore, our decision to evaluate both systems was not redundant but rather strategic and complementary, providing a broader perspective on how different nanostructured carriers can influence the pharmacological performance of 4-PSCO.
References:
- Washington, K.E., Kularatne, R.N., Karmegam, V., Biewer, M.C., Stefan, M.C., 2016. Recent advances in aliphatic polyesters for drug delivery applications. Wiley Interdisciplinary Reviews: Nanomedicine and Nanobiotechnology 9, e1446. https://doi.org/10.1002/wnan.1446
- Espinoza, S.M., Patil, H.I., San Martin Martinez, E., Casañas Pimentel, R., Ige, P.P., 2019. Poly-ε-caprolactone (PCL), a promising polymer for pharmaceutical and biomedical applications: Focus on nanomedicine in cancer. International Journal of Polymeric Materials and Polymeric Biomaterials 69, 85–126. https://doi.org/10.1080/00914037.2018.1539990
- Bhadran, A., Shah, T.K., Babanyinah, G.K., Polara, H., Taslimy, S., Biewer, M.C., Stefan, M.C., 2023. Recent Advances in Polycaprolactones for Anticancer Drug Delivery. Pharmaceutics 15, 1977–1977. https://doi.org/10.3390/pharmaceutics15071977
- Deng, S., Gigliobianco, M. R., Censi, R., Martino, P. D., 2020. Polymeric Nanocapsules as Nanotechnological Alternative for Drug Delivery System: Current Status, Challenges and Opportunities. Nanomaterials 10, 5, 847. https://doi.org/1010.3390/nano10050847
- Pleguezuelos-Villa, M., Nácher, A., Hernández, M.J., Ofelia Vila Buso, M.A., Ruiz Sauri, A., Díez-Sales, O., 2019. Mangiferin nanoemulsions in treatment of inflammatory disorders and skin regeneration. International Journal of Pharmaceutics 564, 299–307. https://doi.org/10.1016/j.ijpharm.2019.04.056.
- Souza, R.L.D., Opretzka, L.C.F., Morais, M.Cd., Melo, Cd.O., Oliveira, B.E.Gd., Sousa, D.Pd., Villarreal, C.F., Oliveira, E.E., 2024. Nanoemulsion Improves the Anti-Inflammatory Effect of Intraperitoneal and Oral Administration of Carvacryl Acetate. Pharmaceuticals 17, 1, 17–17. https://doi.org/10.3390/ph17010017
- Comment - While the introduction gives a general overview of RA pathophysiology and nanotechnology potential, the specific knowledge gap this work addresses could be more explicitly stated. For example, the authors could emphasize the unmet need for RA treatments that simultaneously target joint inflammation, modulate CNS oxidative stress, and address psychiatric comorbidities. Clarifying how current therapeutic and nanocarrier approaches fail to address these aspects would highlight the novelty of their study.
Answer: We thank the reviewer for the relevant comment, with which we fully agree. In the revised manuscript, we will include an additional paragraph to clarify the limitations of current therapeutic and nanocarrier-based approaches for rheumatoid arthritis, thereby highlighting the novelty and relevance of our study (Introduction, page 2).
- Comment - The current transition from RA pathophysiology to nanotechnology applications is somewhat abrupt. Adding a bridging paragraph that connects RA’s biochemical features (e.g., oxidative stress, BBB permeability issues, immune dysregulation) to the rationale for using a lipophilic compound like 4-PSCO and encapsulating it in nanocarriers could improve readability. This would also strengthen the scientific justification for why these formulations may outperform standard delivery methods.
Answer: We appreciate the reviewer’s suggestion and fully agree that a smoother transition would improve the clarity and scientific rationale of the manuscript. In the revised version, we will add a bridging paragraph explicitly linking key biochemical features of RA to the therapeutic potential of 4-PSCO (Introduction session, page 2).

Reviewer 3 Report
Comments and Suggestions for Authors
The manuscript by Caren Aline Ramson Da Fonseca et al. presents a well-structured and comprehensive preclinical study evaluating the efficacy of 4-(phenylselanyl)-2H-chromen-2-one (4-PSCO) in both nanoemulsion (NE) and polymeric nanocapsule (NC) formulations for the treatment of rheumatoid arthritis (RA) and associated comorbidities. The topic is an innovative RA therapies that target pain, inflammation, and neuropsychiatric complications.
I recommend adding a graphical abstract illustrating the proposed mechanism of action of the compounds, highlighting their effects on RA pathology and comorbidities.
Based on the preclinical data presented, is it reasonable to presume that the nanoemulsion and polymeric nanocapsule formulations of 4-PSCO would be safe for human use? Please discuss this point in light of existing toxicological data on organoselenium compounds, cyclodextrins, and the excipients used, and indicate whether additional safety studies are required before translation to clinical settings.
Consider adding a brief perspective on potential clinical translation, including challenges in scaling up the production of the nanocapsules.
These compounds may influence proteins located within lipid rafts. Moreover, agents such as methyl-β-cyclodextrin, which are sometimes used as adjuvants in therapies, are known to solubilize lipid rafts. Could the authors expand the discussion to include the potential role of lipid rafts in their experimental context and the possible interaction with agents like methyl-β-cyclodextrin? I suggest basing this addition on relevant review articles from 2023 and also on the study (Cells. 2022; 11(8): 1288. doi:10.3390/cells11081288), which describes the involvement of lipid rafts and the LRP6 pathway in endothelial cell activation.
Author Response
REVIEWER 3
We would like to thank the reviewer for their suggestions and corrections to the manuscript. We recognize that your comments help improve the quality of the manuscript. The responses to the questions raised are provided in sequence.
Reviewer comments
The manuscript by Caren Aline Ramson Da Fonseca et al. presents a well-structured and comprehensive preclinical study evaluating the efficacy of 4-(phenylselanyl)-2H-chromen-2-one (4-PSCO) in both nanoemulsion (NE) and polymeric nanocapsule (NC) formulations for the treatment of rheumatoid arthritis (RA) and associated comorbidities. The topic is an innovative RA therapies that target pain, inflammation, and neuropsychiatric complications.
- Comment - I recommend adding a graphical abstract illustrating the proposed mechanism of action of the compounds, highlighting their effects on RA pathology and comorbidities.
Answer: We sincerely thank the reviewer for this valuable recommendation. We would like to note that a graphical abstract was already included in the manuscript at the time of submission. This illustration synthesizes and highlights the main effects observed for 4-PSCO, both in its free or nanoencapsulated form, thereby facilitating the visualization and understanding of the central findings of our study.
- Comment - Based on the preclinical data presented, is it reasonable to presume that the nanoemulsion and polymeric nanocapsule formulations of 4-PSCO would be safe for human use? Please discuss this point in light of existing toxicological data on organoselenium compounds, cyclodextrins, and the excipients used, and indicate whether additional safety studies are required before translation to clinical settings.
Answer: We thank the reviewer for this insightful comment. In drug development, safety is a crucial factor. Although selenium compounds exhibit remarkable pharmacological effects, their potential is often limited by toxicity, which may compromise or even invalidate outcomes.
Our preclinical results from the C. elegans survival assay showed that exposure to free 4-PSCO at the highest tested concentration reduced worm survival. The growth of C. elegans is mainly controlled by a well-conserved genetic pathway, making this a reliable measure to assess toxic effects in this organism [1]. However, no significant changes were seen in head thrashes, pharyngeal pumping, or the defecation cycle when animals were exposed to free 4-PSCO. This suggests that the molecule did not impact motility or the neuromuscular system of the worms. Similar results have been reported in the C. elegans model for other organoselenium compounds, supporting a low-toxicity profile [2,3], although some studies have noted adverse effects at lower concentrations [4].
A limitation of the C. elegans model lies in understanding compound toxicokinetics, as nematodes lack both a circulatory system and hepatic first-pass metabolism ([5]. To better assess substance metabolism and administration routes, we conducted an in vivo toxicity assay in mice, which revealed unchanged toxicity parameters. These results, combined with the absence of behavioral alterations in C. elegans and the lack of in vivo toxicological changes, are promising, suggesting a potentially broader therapeutic window for 4-PSCO, particularly in light of the previously reported toxicity of synthetic organoselenium compounds. Recent reviews [6] emphasize that organoselenium compounds may exhibit chronic toxicity related to their thiol-oxidizing properties. Moreover, tissue accumulation and dose- or time-dependent effects are recognized contributors to toxicity. Therefore, while our data point to an encouraging safety signal, they do not exclude potential risks at higher doses or during prolonged exposure.
Regarding the excipients, PCL is FDA-approved, has been extensively studied as a biomaterial, and is regarded as biodegradable, biocompatible, and non-mutagenic [7,8]. The oily core, composed of medium-chain triglycerides, has a long-standing history of pharmaceutical use and a documented low-toxicity profile. Likewise, Span 80 (sorbitan monooleate) and medium-chain triglycerides exhibit a favorable safety profile and are widely employed as surfactants in pharmaceutical formulations. Their use is particularly advantageous due to their high efficiency in solubilizing lipophilic compounds [9,10].
Despite these biocompatibility elements, it would be premature to assume human safety based solely on our preliminary results. For clinical translation, additional Good Laboratory Practice (GLP) and International Council for Harmonisation of Technical Requirements for Pharmaceuticals for Human Use (ICH) guidelines-compliant safety studies are essential. In summary, our findings suggest an initial safety profile that supports the further investigation of 4-PSCO in nanoformulations. Nevertheless, more studies aligned with International Council for Harmonisation of Technical Requirements for Pharmaceuticals for Human Use/Organisation for Economic Cooperation and Development (ICH/OECD) guidelines remain essential before first-in-human studies.
References:
- Hunt, P.R., 2016. The C. elegans model in toxicity testing. Journal of Applied Toxicology 37, 50–59. https://doi.org/10.1002/jat.3357
- Soares, A. T. G., rodrigues, L. B. L., salgueiro, W. G., Dal, F. A.H. C., Rodrigues, C. F., Sacramento, M., Franco, J., Alves, D., Oliveira, R. P., Pinton, S., Ávila, D. S. 2019. Organoselenotriazoles attenuate oxidative damage induced by mitochondrial dysfunction in mev-1 Caenorhabditis elegans mutants. Journal of Trace Elements in Medicine and Biology 53, 34–40. https://doi.org/10.1016/j.jtemb.2019.01.017
- Motta, H. S., Roos, D., Tabarelli, G., Rodrigues, O. E.D., Ávila, D., Quines, C. B., 2020. Activation of SOD-3 is involved in the antioxidant effect of a new class of β-aryl-chalcogenium azide compounds in Caenorhabditis elegans. Anais da Academia Brasileira de Ciências 92, e20181147. https://doi.org/10.1590/0001-3765202020181147
- Avila, D. S., Benedetto, A., Au, C., Manarin, F., Erikson, K., Soares, F. A., Rocha, J. B., Aschner, M., 2012. Organotellurium and organoselenium compounds attenuate Mn-induced toxicity in Caenorhabditis elegans by preventing oxidative stress. Free Radical Biology and Medicine 52, 1903-1910. https://doi.org/10.1016/j.freeradbiomed.2012.02.044
- Weinhouse, C., Truong, L., Meyer, J. N., Allard, P., 2018. Caenorhabditis elegans as an emerging model system in environmental epigenetics. Environmental and Molecular Mutagenesis 59, 560-575. https://doi.org/10.1002/em.22203
- Nogueira, C.W., Barbosa, N. V., Rocha, J. B. T., 2022. Toxicology and pharmacology of synthetic organoselenium compounds: an update. Archives of Toxicology 95, 1179-1226. https://doi.org/10.1007/s00204-021-03003-5
- Bhadran, A., Shah, T.K., Babanyinah, G.K., Polara, H., Taslimy, S., Biewer, M.C., Stefan, M.C., 2023. Recent Advances in Polycaprolactones for Anticancer Drug Delivery. Pharmaceutics 15, 1977–1977. https://doi.org/10.3390/pharmaceutics15071977
- Christodoulou, E., Notopoulou, M., Nakiou, E., Kostoglou, M., Barmpalexis, P., Bikiaris, D. N., 2022. Branched Poly(ε-caprolactone)-Based Copolyesters of Different Architectures and Their Use in the Preparation of Anticancer Drug-Loaded Nanoparticles. International Journal of Molecular Sciences 23, 15393. https://doi.org/10.3390/ijms232315393
- Zapolski, R., Gasztych, M., Jastrząb-Miśkiewicz, B., Jankowska-Konsur, A., Musiał, W., 2025. The Properties of the Monolayers of Sorbitan Lipids as Informative Factors on the Hydrophilic-Lipophilic Balance Value of Their Mixtures, Proposed for Dermatological Applications. Molecules 30, 1841. https://doi.org/10.3390/molecules30081841
- Deng, S., Gigliobianco, M. R., Censi, R., Di Martino, P., 2020. Polymeric Nanocapsules as Nanotechnological Alternative for Drug Delivery System: Current Status, Challenges and Opportunities. Nanomaterials 10, 847. https://doi.org/10.3390/nano10050847
- Comment - Consider adding a brief perspective on potential clinical translation, including challenges in scaling up the production of the nanocapsules.
Answer: We appreciate the reviewer's valuable suggestion. A discussion on the potential clinical translation, including the scale-up of nanocapsule production, has been added to the final manuscript (Future Perspectives section, page 24).
- Comment - These compounds may influence proteins located within lipid rafts. Moreover, agents such as methyl-β-cyclodextrin, which are sometimes used as adjuvants in therapies, are known to solubilize lipid rafts. Could the authors expand the discussion to include the potential role of lipid rafts in their experimental context and the possible interaction with agents like methyl-β-cyclodextrin? I suggest basing this addition on relevant review articles from 2023 and also on the study (Cells. 2022; 11(8): 1288. doi:10.3390/cells11081288), which describes the involvement of lipid rafts and the LRP6 pathway in endothelial cell activation.
Answer: We thank the reviewer for their feedback. In response, we have thoroughly revised the Discussion section to provide a more detailed analysis, emphasizing the potential role of lipid rafts within our experimental context. To address the reviewer's suggestion, additional content has been included to further clarify and highlight this aspect (Discussion section, page 15).

Round 2
Reviewer 1 Report
Comments and Suggestions for Authors
I re-reviewed the manuscript entitled “Preclinical Evaluation of Nanoemulsion and Polymeric Nanocapsule Delivery Systems of 4-(Phenylselanyl)-2H-chromen-2-one for Rheumatoid Arthritis and Comorbidities”. The authors have performed fairly well in revising the manuscript and improving its quality. However, in some cases the suggested revisions were insufficiently done. Also, some new questions arose about the provided revisions. The following issues should be addressed to make the manuscript of appropriate quality for publication in Pharmaceuticals.
In the introduction, the authors stated nanotechnology emerging discipline, so provide some examples of Nanoemulsion and Polymeric Nanocapsule Delivery Systems for RA with their limitations by highlighting the authors' work.
Please explain why the author chose to use Nanoemulsion and Polymeric Nanocapsule Delivery Systems instead of Polymeric Nanoparticles.
The rationale for developing this specific delivery system for 4-(Phenylselanyl)-2H-chromen-2-one could be stated more clearly upfront. Why was this particular approach chosen over other potential delivery methods?
A more thorough discussion of potential limitations of the approach and study design would provide a more balanced perspective. The paper could benefit from a clearer discussion of next steps and future research directions to build on this work.
The in vivo studies were relatively short-term. Longer-term studies would be beneficial to assess the sustained efficacy and potential side effects of prolonged use of the Nanoemulsion and Polymeric Nanocapsule Delivery Systems of 4-(Phenylselanyl)-2H-chromen-2-one.
The study doesn't explore different dosing regimens. Optimizing the dose could potentially improve efficacy or reduce side effects.
The study uses only one type of animal model (CFA-induced arthritis in mice). Including other animal models of rheumatoid arthritis could strengthen the findings.
Discussion on the potential for scaling up the production of these nanocarriers for commercial use is lacking.
I suggest that you consider incorporating these recent articles to provide a more comprehensive view of the current state of the field and to highlight the latest innovations in nanocrystal drug delivery against RA model in the in vivo methodology section regarding behavioral parameters and biochemical assay https://doi.org/10.1016/j.bioadv.2024.214093
Author Response
REVIEWER 1
We would like to thank the reviewer for their suggestions and corrections to the manuscript. We recognize that your comments significantly contribute to improving the quality of the manuscript. The responses to the questions raised are given in sequence.
Reviewer comment:
I re-reviewed the manuscript entitled “Preclinical Evaluation of Nanoemulsion and Polymeric Nanocapsule Delivery Systems of 4-(Phenylselanyl)-2H-chromen-2-one for Rheumatoid Arthritis and Comorbidities”. The authors have performed fairly well in revising the manuscript and improving its quality. However, in some cases the suggested revisions were insufficiently done. Also, some new questions arose about the provided revisions. The following issues should be addressed to make the manuscript of appropriate quality for publication in Pharmaceuticals.
- Comment - In the introduction, the authors stated nanotechnology emerging discipline, so provide some examples of Nanoemulsion and Polymeric Nanocapsule Delivery Systems for RA with their limitations by highlighting the authors' work.
Answer: We thank the reviewer for this comment. In the Introduction, our focus was to highlight the advantages of polymeric nanoemulsions and nanocapsules, along with the outcomes of our previous study with 4-(Phenylselanyl)-2H-chromen-2-one (4-PSCO), in which the compound was evaluated both in its free and nanoencapsulated forms. We considered this particularly relevant, as it established the foundation for the present work, from which we refined the treatment regimen and duration to advance toward a safer and more effective therapeutic strategy for rheumatoid arthritis (RA) (Introduction section, Pages 2-3).
Additionally, in the Introduction, we emphasized the main advantages of nanocarriers, including improved bioavailability, pharmacokinetics, pharmacodynamics, therapeutic efficacy, and site-specific delivery, underscoring the importance of nanotechnology in the development of our drug candidates (Introduction, Pages 2-3). Regarding the limitations of nanocarrier systems for RA, we agree that these are valuable to contextualize our work. Therefore, these aspects were addressed and expanded in the Discussion section, where we compared the nanocarriers used in this study with alternative delivery strategies reported in the literature, thereby highlighting both the novelty and relevance of our approach (Discussion section, Page 14).
- Comment - Please explain why the author chose to use Nanoemulsion and Polymeric Nanocapsule Delivery Systems instead of Polymeric Nanoparticles.
Answer: We appreciate the reviewer’s question. The choice of nanoemulsion and polymeric nanocapsule formulations was guided by the physicochemical characteristics of 4-PSCO, a highly lipophilic compound. Both nanoemulsions and polymeric nanocapsules improve the solubility of lipophilic molecules, thereby enhancing their bioavailability and absorption. In addition, due to their polymeric shell, polymeric nanocapsules provide superior control over drug release compared to nanoemulsions [1]. This comparison enables us to more accurately evaluate the impact of release control in the enhancement of pharmacological effects when compared to the free form.
References:
- Cardoso, A.M., de Oliveira, E.G., Coradini, K., Bruinsmann, F.A., Aguirre. T., Lorenzoni, R., Barcelos, R.C.S., Roversi, K., Rossato, D.R., Pohlmann, A.R., Guterres, S.S., Burger, M.E., Beck, R.C.R., 2019. Chitosan hydrogels containing nanoencapsulated phenytoin for cutaneous use: Skin permeation/penetration and efficacy in wound healing. Materials Science and Engineering: C 96, 205-217. https://doi.org/10.1016/j.msec.2018.11.013
- Comment - The rationale for developing this specific delivery system for 4-(Phenylselanyl)-2H-chromen-2-one could be stated more clearly upfront. Why was this particular approach chosen over other potential delivery methods?
Answer: We thank the reviewer for bringing this important point to our attention. Polymeric nanocapsules consist of nanoparticles with an oily core encased in a thin polymeric shell, while nanoemulsions are oil droplets dispersed in an aqueous phase, stabilized by surfactants. In our previous study [1], we developed a polymeric nanocapsule using ethylcellulose polymer, a hydrophobic cellulose derivative known for its high controllability in drug release [2]. Although nanoencapsulated 4-PSCO showed superior pharmacological performance compared to its free form in specific parameters, we hypothesized that modifying the qualitative and quantitative formulation’s composition could further enhance its therapeutic effects when compared to free form. Therefore, in the present study, we chose to develop two distinct nanostructured systems: a polymeric nanocapsule using Poly(ε-caprolactone) polymer and a nanoemulsion. To address the reviewer´s comments, we added this information to the revised version of the manuscript (Introduction section, Pages 2-3).
References:
- da Fonseca, C.A.R., Prado, V.C., Paltian, J.J., Kazmierczak, J.C., Schumacher, R.F., Sari, M.H.M., Cordeiro, L.M., da Silva, A.F., Soares, F.A.A., Oliboni, R.D.S., Luchese, C., Cruz, L., Wilhelm, E.A., 2024. 4-(Phenylselanyl)-2H-chromen-2-one-Loaded Nanocapsule Suspension-A Promising Breakthrough in Pain Management: Comprehensive Molecular Docking, Formulation Design, and Toxicological and Pharmacological Assessments in Mice. Pharmaceutics 16, 269. https://doi.org/10.3390/pharmaceutics16020269
- Adeleke, O.A., 2019. Premium ethylcellulose polymer based architectures at work in drug delivery. International Journal of Pharmaceutics: X 1, 100023. https:// doi.org/10.1016/j.ijpx.2019.100023
- Comment - A more thorough discussion of potential limitations of the approach and study design would provide a more balanced perspective. The paper could benefit from a clearer discussion of next steps and future research directions to build on this work.
Answer: We thank the reviewer for this important comment. We agree that a more detailed discussion of the limitations of our study enriches the manuscript and provides a more balanced perspective. Accordingly, as suggested, we have expanded the Limitations section to include important considerations such as the relatively short duration of the in vivo experiments, the use of a single animal model, and the absence of dosing regimen optimization (Limitations section, Pages 23-24).
Furthermore, we have included additional information in a section outlining next steps and future research directions. These include conducting long-term studies to assess the sustained efficacy and safety of 4-PSCO nanoformulations, exploring different dosing strategies to optimize therapeutic outcomes, and evaluating the effects of treatment in additional preclinical models of RA. These additions strengthen the manuscript by clarifying both the current scope of our work and the path for future investigations (Future Perspectives section, Page 24).
- Comment - The in vivo studies were relatively short-term. Longer-term studies would be beneficial to assess the sustained efficacy and potential side effects of prolonged use of the Nanoemulsion and Polymeric Nanocapsule Delivery Systems of 4-(Phenylselanyl)-2H-chromen-2-one.
Answer: We appreciate the reviewer’s thoughtful observation. Indeed, the in vivo experiments conducted in this study were designed as short-term protocols to evaluate the therapeutic efficacy and safety of 4-PSCO in nanoemulsion and polymeric nanocapsule formulations. We fully agree that longer-term studies are essential to determine sustained efficacy, pharmacokinetic behavior, and potential side effects associated with chronic administration. Such investigations are part of our future research plans and will be critical to advance the translational potential of these nanostructured delivery systems for RA (Future Perspectives section, Page 24).
- Comment - The study doesn't explore different dosing regimens. Optimizing the dose could potentially improve efficacy or reduce side effects.
Answer: We thank the reviewer for this comment and agree with the observation. Optimizing dosing regimens is indeed a crucial step and represents one of the primary objectives of our ongoing study. In this follow-up work, we are evaluating an intermittent dosing protocol to understand better the therapeutic effects of both free and nanoencapsulated 4-PSCO. Additionally, we are investigating sex-related differences in RA and assessing motor function to broaden the translational relevance of our findings.
At the same time, we acknowledge the importance of continuous dosing regimens, as explored in the present study, since they provide critical information regarding potential toxicity and safety under prolonged administration. Notably, in both previous and current experiments, neither free, nanoencapsulated, nor nanoemulsified 4-PSCO demonstrated unacceptable toxicity or adverse effects, thereby reinforcing the compound’s safety profile.
Furthermore, in our group’s initial study evaluating the anti-edematogenic, anti-inflammatory, and analgesic effects of 4-PSCO in well-established experimental models of pain and inflammation, different doses (1 and 5 mg/kg) were tested. Based on the relevant findings obtained with the 1 mg/kg dose, as well as evidence from studies involving other selenium-based organoselenium compounds and selenocoumarins, we opted to continue using this low dose in subsequent studies, considering its therapeutic safety margin.
Notably, the present study offers significant advantages by establishing a robust proof-of-concept that validates the therapeutic potential of 4-PSCO in both free and nanocarrier forms. Beyond confirming its safety, our data highlight its ability to act on both peripheral and central mechanisms relevant to RA, thereby advancing a multitarget therapeutic approach. These findings not only strengthen the translational value of 4-PSCO but also provide a solid foundation for future studies focused on optimizing dosage regimens to maximize clinical applicability and tolerability.
However, to address the reviewer’s concern, we have included in the Limitations section the consideration that investigating alternative dosing strategies, including interval-based regimens, could further optimize therapeutic outcomes and strengthen the clinical relevance of our findings (Limitations section, Pages 23-24).
- Comment - The study uses only one type of animal model (CFA-induced arthritis in mice). Including other animal models of rheumatoid arthritis could strengthen the findings.
Answer: We thank and agree with the reviewer’s comment. Indeed, the inclusion of additional RA models could further strengthen our findings. However, the CFA-induced arthritis model remains one of the most widely employed experimental approaches, as it reliably reproduces key clinical and pathological features of human RA and is therefore considered a standard in preclinical pharmacological research for evaluating the efficacy of novel anti-inflammatory and analgesic agents [1,2].
CFA administration triggers a progressive inflammatory response, initially characterized by local erythema and edema peaking within 6–8 hours and lasting up to 4 days. This acute phase is followed by systemic increases in inflammatory markers and the development of joint hyperalgesia and edema within 1–2 weeks, associated with neutrophil infiltration and synovial proliferation. The sustained release of pro-inflammatory cytokines, particularly TNF-α and IL-1β, amplifies the inflammatory cascade by promoting cellular adhesion, acute-phase protein synthesis, and tissue degradation, ultimately leading to chronic arthritis through positive feedback mechanisms [3,4].
Thus, the chosen model effectively mimics hallmark signs and symptoms of RA, offering both practicality and reproducibility in disease induction. Furthermore, in line with the ethical principles of the 3Rs (Replacement, Reduction, and Refinement), additional models were not included to minimize animal use while still ensuring robust and clinically relevant findings.
However, to address the reviewer’s concern, we have included in the Limitations section the consideration of complementary models, such as collagen-induced arthritis and genetically modified animals, which could provide a more comprehensive understanding of the disease and enhance the impact of our findings (Limitations section, Pages 23-24).
References:
- Ahmed, E.A., Ahmed, O.M., Fahim, H.I., Ali, T.M., Elesawy, B.H., Ashour, M.B., 2021. Potency of Bone Marrow-Derived Mesenchymal Stem Cells and Indomethacin in Complete Freund’s Adjuvant-Induced Arthritic Rats: Roles of TNF-α, IL-10, iNOS, MMP-9, and TGF-β1. Stem Cells International 2021, 1–11. https://doi.org/10.1155/2021/6665601
- Lal, R., Dhaliwal, J., Dhaliwal, N., Dharavath, R.N., Chopra, K., 2021. Activation of the Nrf2/HO-1 signaling pathway by dimethyl fumarate ameliorates complete Freund’s adjuvant-induced arthritis in rats. European Journal of Pharmacology 899, 174044. https://doi.org/10.1016/j.ejphar.2021.174044
- Nasuti, C., Fedeli, D., Bordoni, L., Piangerelli, M., Servili, M., Selvaggini, R., Gabbianelli, R., 2019. Anti-Inflammatory, Anti-Arthritic and Anti-Nociceptive Activities of Nigella sativa Oil in a Rat Model of Arthritis. Antioxidants 8, 342. https://doi.org/10.3390/antiox8090342
- Noh, A.S.M., Chuan, T.D., Khir, N.A.M., Zin, A.A.M., Ghazali, A.K., Long, I., Ab Aziz, C.B., Ismail, C.A.N., 2021. Effects of different doses of complete Freund's adjuvant on nociceptive behaviour and inflammatory parameters in polyarthritic rat model mimicking rheumatoid arthritis. PLoS One 16, e0260423. https://doi.org/10.1371/journal.pone.0260423
- Comment - Discussion on the potential for scaling up the production of these nanocarriers for commercial use is lacking.
Answer: We thank the reviewer for this comment. We acknowledge that the scalability of nanocarrier production is a critical step toward clinical and commercial translation. While the present study focused primarily on preclinical evaluation, we have expanded the Discussion to address this critical aspect. In particular, we emphasize that large-scale manufacturing of polymeric nanocapsules and nanoemulsions requires rigorous control of key parameters, including particle size distribution, encapsulation efficiency, and physicochemical stability to ensure reproducibility. Moreover, we emphasize that challenges related to sterilization, endotoxin removal, and strict adherence to Good Manufacturing Practices are crucial for advancing these systems toward clinical-grade formulations (Discussion section, Page 15). These considerations have also been incorporated into the Future Perspectives section (Future Perspectives section, Page 24).
- Comment - I suggest that you consider incorporating these recent articles to provide a more comprehensive view of the current state of the field and to highlight the latest innovations in nanocrystal drug delivery against the RA model in the in vivo methodology section regarding behavioral parameters and biochemical assay https://doi.org/10.1016/j.bioadv.2024.214093
Answer: We thank the reviewer for the suggestion. The recommended article by Khan et al. (2024) has been incorporated into the Discussion section to provide a more comprehensive view of recent advances in nanomicelle drug delivery strategies for rheumatoid arthritis. (Discussion section, Page 14).

Reviewer 2 Report
Comments and Suggestions for Authors
The authors address all my questions very well, and the article can be accepted in its current format.
Author Response
Thank you for your positive assessment. We’re pleased our revisions fully addressed your questions and appreciate your recommendation for acceptance. We remain available for any final editorial adjustments if needed.
Reviewer 3 Report
Comments and Suggestions for Authors
The authors have completed my requests
Author Response

(The authors gave the same response as above.)
